# Acute hunger does not always undermine prosociality

Jan A. Häusser[1,9]*, Christina Stahlecker [1,9]*, Andreas Mojzisch[2], Johannes Leder [3]
Paul A.M. Van Lange[4,5] & Nadira S. Faber [6,7,8]

It has been argued that, when they are acutely hungry, people act in self-protective ways by keeping resources to themselves rather than sharing them. In four studies, using experimental, quasi-experimental, and correlational designs (total N = 795), we examine the effects of acute hunger on prosociality in a wide variety of non-interdependent tasks (e.g. dictator game) and interdependent tasks (e.g. public goods games). While our procedures successfully increase subjective hunger and decrease blood glucose, we do not find significant effects of hunger on prosociality. This is true for both decisions incentivized with money and with food. Meta-analysis across all tasks reveals a very small effect of hunger on prosociality in non-interdependent tasks (d = 0.108), and a non-significant effect in interdependent tasks (d = −0.076). In study five (N = 197), we show that, in stark contrast to our empirical findings, people hold strong lay theories that hunger undermines prosociality.

[1] Department of Psychology, Justus-Liebig-University Gießen, Otto-Behaghel-Str. 10D, 35394 Gießen, Germany. [2] Department of Psychology, University of Hildesheim, Universitätsplatz 1, 31141 Hildesheim, Germany. [3] Department of Psychology, University of Bamberg, Markusplatz 3, 96047 Bamberg, Germany. [4] Department of Experimental and Applied Psychology, VU Amsterdam, Van der Boechorststraat 7, 1081 BT Amsterdam, The Netherlands. [5] Institute for Brain and Behavior Amsterdam (IBBA), Van der Boechorststraat 7, 1081 BT Amsterdam, The Netherlands. [6] Department of Experimental Psychology, University of Oxford, New Radcliffe House, Radcliffe Observatory Quarter, Woodstock Road, Oxford OX2 6GG, UK. [7] Oxford Uehiro Centre for Practical Ethics, University of Oxford, Suite 8, Littlegate House, St Ebbe's Street, Oxford OX1 1PT, UK. [8] College of Life and Environmental Sciences, University of Exeter, Washington Singer Building, Exeter EX4 4QG, UK. [9] These authors contributed equally: Jan A. Häusser, Christina Stahlecker.
*email: Jan.A.Haeusser@psychol.uni-giessen.de; Christina.Stahlecker@psychol.uni-giessen.de

D o people become less prosocial when they are hungry? Do they share less? Various theories favor the idea that facing limited resources, or the thwarting of basic needs such as food to satisfy hunger, brings about a focus on immediate self-interest[1–4]. For example, conservation of resources theory[5,6] argues that when facing threats, people tend to focus on themselves and save their resources. While the idea is compelling, does acute hunger indeed make people less helpful and cooperative? In this paper, we address this fundamental question in both non-interdependent and interdependent settings. Acute hunger due to temporary deprivation of food is characterized by craving for food, feeling hungry, and, on a physiological level, decreased blood glucose levels.

It has been argued that hunger, as a signal of limited resource availability, reduces prosociality, that is, the willingness to invest one's own resources (e.g., money, time, effort) to help others[7,8]. Such "pure prosociality" is typically measured in non-interdependent settings. In such settings, the outcome an actor A gets exclusively depends on A's own unilateral decisions, for example, how many resources to allocate to another person or a group. A's outcome, the consequences of the decision, does not depend on the choices of other agents, and hence, A's behavior is not subject to strategic considerations like concerns over reciprocity. Donating money to a noble cause[9] is a prototypical example of prosocial behavior in non-interdependent settings.

One of the most prominent experimental paradigms for non-interdependent settings is the dictator game[10] (DG). In the DG, a decision maker (i.e., the dictator) receives an endowment, and has to decide how to split this endowment between herself and another anonymous participant (i.e., the recipient). Because the recipient is powerless, the situation is non-interdependent, that is, the payoff the dictator receives is only dependent on the split of the endowment that she suggested. As expectations of reciprocity do not play a role in the DG, the dictator's behavior is a measure of pure prosociality[11,12]. What factors influence the decision to keep versus share resources in such non-interdependent settings? As a relevant psycho-physiological influence, hunger might bring about a focus on the immediate self-interest and might, therefore, reduce prosociality in non-interdependent settings. Some studies have started to address this question. Indeed, there is preliminary evidence in support of the idea that acute hunger increases selfishness[13–15].

In one study[15], food deprivation reduced participants' intention to donate money to charity, and in another study[13], food abundance, induced by participants' consumption of an energy bar prior to their decisions, resulted in a stronger intention to donate money to charity. In line with this, there is also tentative evidence that increased blood glucose levels lead to higher contributions in the DG[14]. The picture, however, is somewhat inconsistent, since two other studies found no significant effect of experimentally manipulated hunger on charitable giving[16] or experimentally manipulated blood glucose levels on the amount of money participants shared in a DG[17]. A closer look at the latter null effect, however, revealed that it was due to two countervailing effects of blood glucose on prosociality[17]. A marginally significant direct effect of blood glucose on the amount of money shared in the DG is in line with the idea of resource accumulation and consistent with the previous findings. At the same time, blood glucose levels were also negatively related to support for social welfare, which, in turn, was positively related to sharing behavior. Hence, via this indirect effect, decreased blood glucose may have led to increased prosociality. This finding could be interpreted in terms of low blood glucose levels increasing selfish tendencies, but these are canceled out by maintenance or even strengthening of prosocial norms.

As summarized above, hunger can reduce prosociality in non-interdependent settings, though the evidence should be considered tentative rather than conclusive. Does hunger also undermine prosociality in interdependent settings? In such settings, the outcome an actor A gets does not exclusively depend on A's unilateral decision but depends on the choices of other agents. Hence, when they are mindful of their own outcomes, in interdependent situations, individuals cannot simply monitor only their own behavior, but have to consider key aspects of the social context, for example, their beliefs about the choices other people will make. Hence, unlike non-interdependent settings, interdependent settings do not measure pure prosociality as the decisions of an actor A can be influenced by other motives, such as strategic concerns. A prominent experimental paradigm for interdependent settings is the Ultimatum Game (UG)[18]. The UG is an experimental game with two players. One player, the proposer, receives an endowment and she has to decide how to split this endowment between herself and the second player, the responder. The responder, in turn, has to decide whether she accepts the offer or rejects it. If the offer is accepted, the money is paid out accordingly. In case of rejection, both players receive nothing. As opposed to the DG, where the recipient of the endowment is powerless, in the UG, the proposer has to anticipate the reaction of the responder. Hence, a fair offer of the proposer in the UG can be motivated either by a fairness motive (i.e., a prosocial motive) or by the fear of an unfair offer being rejected. (For an overview of a variety of non-interdependent and interdependent tasks and their payoff structures, see Fehr & Schmidt[19]; Kelley et al.[20]). In interdependent settings, the choice whether to act prosocially or not also entails strategic considerations and beliefs about the other person's response[20–22]. The effects of hunger on decisions in interdependent settings could, therefore, be the result of hunger effects on prosociality but also of hunger effects on strategic decision making. We have argued previously that humans can compensate for their own psycho-physiological impairments when the social context requires them to[23]. Hence, it is possible that, even if hunger enhances selfish tendencies, those will not necessarily translate into decreased prosociality in interdependent settings due to strategic concerns that also influence the decision.

To the best of our knowledge, only one study[16] has investigated the effects of hunger in interdependent settings. In this study, Rantapuska and colleagues investigated the effects of experimentally manipulated hunger in two cooperation paradigms. Their study yielded results that were somewhat inconclusive, with increased prosociality in one of the tasks and a null finding in the other one, thus emphasizing the need for further research.

Our research was set out to test the effects of hunger on prosociality using a variety of study designs, different operationalizations of hunger, and different tasks. We conducted four studies, two of them preregistered, investigating the effects of acute hunger and blood glucose levels in non-interdependent and interdependent settings. Two studies were laboratory experiments, the other two were field studies, using a correlational and a quasi-experimental approach (Table 1 provides an overview of the study designs, samples and measures). In a fifth study, we examined lay theories about the effects of acute hunger on prosociality.

As non-interdependent measures we used the DG, social value orientation[24–26], social mindfulness[27], and a volunteering task[28]. Social value orientation (SVO) represents a person's preference for (hypothetical) distributions of money between herself and another person in a set of non-constant-sum DGs[29,30]. More precisely, participants have to decide on a series of DG-like decisions, with different endowments and different distribution options. The SVO measure extends rational self-interest by

**Table 1 Overview of the methods used in Studies 1–4, including samples, design, and predictor variables, as well as dependent variables (DVs), either in non-interdependent or interdependent tasks**

| Method | Study 1 | Study 2 | Study 3 | Study 4 |
|---|---|---|---|---|
| Sample size | $N = 62$ | $N = 103$ | $N = 267$ | $N = 363$ |
| Setting | Laboratory | Laboratory | Field | Field |
| Design | Experimental | Experimental | Correlational | Quasi-Experimental |
| Predictors | Induced hunger | Induced hunger | Natural hunger | Natural hunger (before vs. after lunch), Incentive (food or money) |
| Manipulation Check | Subjective hunger, blood glucose | Subjective hunger, blood glucose | – | Subjective hunger |
| DVs_non-interdependent | – | SVO, social mindfulness | SVO | SVO, DG, volunteering |
| DVs_interdependent | PGG, SHG | PGG, UG | – | – |

simultaneously measuring the value people assign to other people's outcomes[25,31]. SVO has been validated to be predictive of real-life prosocial behavior, such as donations to noble causes[9,32], and volunteering[28,33], as well as costly cooperation in economic games[34–37]. Social mindfulness[27] refers to the extent to which an individual's decisions leave other people in decisional control, thereby respecting other people's interest to choose freely for themselves. Behaving socially mindful can, therefore, be understood as a prosocial act, as it ensures options for the other person rather than removing options. It has been conceptualized as "low-cost cooperation"[27,38]. Volunteering[28,33] encompasses the investment of time resources to benefit others.

As interdependent tasks, we used the UG, the Public Goods Game[39,40], and the Stag Hunt Game[41]. In the Public Goods Game (PGG), multiple players can decide whether or not to contribute to a common pool that is afterwards multiplied by a constant and then split equally among all players. The prosocial choice is to contribute everything to the common pool as it increases the joint outcome. However, freeriding is possible, as the individual outcome can be maximized by keeping everything for oneself and still profiting from others' contributions to the common pool[40]. In the Stag Hunt Game (SHG), two players simultaneously decide between a cooperative, high pay-off, but socially risky option ('hunting a stag') and an uncooperative, low-pay-off, but safe option ('hunting a hare'). Hunting the stag, the prosocial choice, results in a higher pay-off, but only if both players decide to hunt the stag. The hare, in contrast, can be hunted down independent of the decisions of the other player, but results in a lower payoff.

In four studies (total $N = 795$), using a variety of non-interdependent and interdependent tasks, and different study designs (see Table 1), we investigate the effects of acute hunger on prosociality. While our (quasi-) experimental manipulations of hunger are successful and strong (i.e., increased feelings of acute hunger and decreased levels of blood glucose), we do not find significant effects of hunger on prosociality in the individual DVs. This is true even for decisions incentivized with food rather than money. Meta-analyses across the different tasks show only a very small overall effect of hunger on prosociality in non-interdependent tasks ($d = 0.108$), and a non-significant effect in interdependent tasks ($d = -0.076$). In a fifth study ($N = 197$), we find that, in stark contrast to our empirical results, lay people hold the belief that acute hunger does undermine prosociality. Hence, while the idea that hunger decreases prosociality seems compelling, and there has been prior evidence in support of it, our results indicate that this effect is very weak at best. Especially in social contexts that convey interdependence, people seem to respond to the actual or perceived social requirements of the situation. In conclusion, we suggest that hunger often does not translate into more selfishness because many situations share some elements of interdependence — when other people notice our actions and are able to respond to them.

## Laboratory experiments 1 and 2

In Studies 1 and 2, we experimentally manipulated acute hunger. Participants were randomly assigned to either a hunger condition or a control condition. All participants were instructed not to eat anything after 10 pm the previous night until the experimental session the following day. Participants in the control condition then received food before completing the tasks. Self-reported hunger as well as blood glucose levels at baseline ($t1$) and before the tasks (i.e., after food consumption of the control group; $t2$), were used as manipulation checks. Study 1 ($N = 62$) was a first experimental exploration of the effects of hunger in interdependent settings, in which we used a PGG and a SHG.

To rule out the possibility that, in Study 1, effects were too small to be detected with our sample size, or that the manipulation of hunger and blood glucose was not strong enough, in Study 2, we used a stronger manipulation of hunger (i.e., experimental sessions were scheduled later, and participants in the control group received a larger amount of sugar) and a larger sample ($N = 103$). Again, we used a PGG and, in addition, a UG. We also included measures of non-interdependent prosociality, namely SVO, and social mindfulness (see Table 1 for an overview of the study designs, samples and measures).

## Results Study 1 and 2

**Study 1 manipulation check.** Repeated-measures ANOVAs showed significant interactions between experimental condition and time of measurement for both self-reported hunger and blood glucose levels (subjective hunger: $F(1, 60) = 38.41$, $p < 0.001$, $\eta_p^2 = 0.40$; blood glucose: $F(1, 60) = 8.15$, $p = 0.006$, $\eta_p^2 = 0.13$). In the control condition, subjective hunger decreased ($M_{t1} = 4.30$, $SD_{t1} = 2.25$, $M_{t2} = 1.67$, $SD_{t2} = 1.89$), $t(60) = 7.11$, $p < 0.001$, $d = -1.34$, and blood glucose levels increased ($M_{t1} = 4.38$ mmol/l, $SD_{t1} = 0.86$, $M_{t2} = 5.10$ mmol/l, $SD_{t2} = 1.09$), $t(60) = 2.88$, $p = 0.007$, $d = 0.54$. As expected, we found no changes in blood glucose and subjective hunger in the hunger condition (all $ps > 0.25$). At $t2$, subjective hunger was significantly higher ($p < 0.001$, $d = 1.76$) and blood glucose was significantly lower ($p = 0.044$, $d = 0.47$) in the hunger condition (subjective hunger: $M_{t2} = 5.09$, $SD_{t2} = 2.00$; blood glucose: $M_{t2} = 4.59$, $SD_{t2} = 0.64$), as compared to the control condition.

**Study 1 public goods game.** The amount of money participants contributed to the common pool in the PGG did not differ significantly between the hunger condition ($M = 4.52$, $SD = 2.08$) and the control condition ($M = 3.74$, $SD = 2.22$), $t(60) = 1.42$, $p = 0.162$, $d = -0.37$.

**Study 1 stag hunt game.** In the hunger condition, 30 out of 31 participants (97%) chose the cooperative option (hunt the stag), whereas in the control condition, 25 out of 31 participants (81%)

cooperated, $\chi^2(1, N = 62) = 4.03$, $p = 0.045$, $\varphi = -0.26$ (n.s. due to adjusted $\alpha = 0.017$, see method section for details).

**Study 2 manipulation check.** Repeated-measures ANOVAs revealed interactions between experimental condition and time of measurement for both self-reported hunger and blood glucose levels (subjective hunger: $F(1, 100) = 75.85$, $p < 0.001$, $\eta_p^2 = 0.43$; blood glucose: $F(1, 98) = 91.06$, $p < 0.001$, $\eta_p^2 = 0.48$). In the control condition, self-reported hunger decreased ($M_{t1} = 7.40$, $SD_{t1} = 1.71$, $M_{t2} = 4.42$, $SD_{t2} = 2.17$), $t(100) = 9.88$, $p < 0.001$, $d = -1.43$, and blood glucose levels increased ($M_{t1} = 4.73$ mmol/l, $SD_{t1} = 1.13$, $M_{t2} = 7.45$ mmol/l, $SD_{t2} = 1.91$), $t(98) = 10.41$, $p < 0.001$, $d = 1.34$. As in Study 1, we found no changes in blood glucose and subjective hunger in the hunger condition (all $ps > 0.2$). At $t2$, subjective hunger was significantly higher ($p < 0.001$, $d = 1.41$), and blood glucose was significantly lower ($p = 0.001$, $d = 1.80$) in the hunger condition (subjective hunger: $M_{t2} = 7.35$, $SD_{t2} = 1.99$; blood glucose: $M_{t2} = 4.94$, $SD_{t2} = 0.57$), as compared to the control condition.

**Study 2 public goods game.** Contributions to the common pool did not differ between the hunger condition ($M = 5.73$, $SD = 2.99$) and the control condition ($M = 6.04$, $SD = 2.90$), $t(100) = 0.53$, $p = 0.597$, $d = 0.11$.

**Study 2 ultimatum game.** There was no significant effect of experimental condition on the amounts of money participants offered to the responder in the UG ($M_{hunger} = 4.49$, $SD_{hunger} = 1.49$; $M_{control} = 4.76$, $SD_{control} = 1.26$), $t(94) = 0.941$, $p = 0.349$, $d = 0.20$.

**Study 2 social value orientation.** Due to computer glitches, SVO values are missing for seven participants. We used an ANCOVA with SVO angle[25] as dependent variable and baseline-SVO as covariate. We found that hungry participants were not significantly lower in SVO ($M = 37.14$, $SD = 8.74$) than participants in the control condition ($M = 40.43$, $SD = 4.29$), $(1,95) = 3.99$, $p = 0.049$, $\eta_p^2 = 0.04$ (Bonferroni-corrected $\alpha$-level is 0.008).

**Study 2 social mindfulness.** We found no significant difference in ratios of socially mindful choices in the hunger condition ($M = 0.73$, $SD = 0.16$) as compared to the control condition ($M = 0.77$, $SD = 0.16$), $F(1,101) = 1.63$, $p = 0.204$, $d = 0.25$.

### Correlational and quasi-experimental field Studies 3 and 4

To increase our confidence in the null findings in Studies 1 and 2, we decided to conduct two additional pre-registered, highly powered studies in naturalistic settings. In these studies, we focused on non-interdependent prosocial behavior for two reasons. First, hunger-induced selfishness should most likely manifest in resource allocation or sharing situations, where it is easy to give in to selfish tendencies and keep everything for oneself, as this would have no social consequences. If acute hunger does not affect prosocial behavior in such non-interdependent situations, it is less plausible that it will in interdependent situations that involve social control. Second, for non-interdependent tasks, there is at least some tentative empirical evidence suggesting negative effects of hunger on prosociality, whereas there is basically no conclusive evidence for interdependent tasks.

Therefore, in Study 3 ($N = 267$), we further investigated the relationship between subjective hunger and SVO in a correlational field study. Students completed a short paper-and-pencil survey as part of their lecture at a German university. In Study 4 ($N = 363$), we used a quasi-experimental approach by recruiting participants

in front of a cafeteria right before lunch (hungry) or after lunch (control). We assessed the effects of hunger on different non-interdependent measures of prosociality: a measure of SVO, a DG, and a volunteering task, in which participants indicated their willingness to participate in a future study without receiving financial compensation, and the time they were willing to spend. Subjective hunger was measured as a manipulation check.

We also examined whether a potential effect of hunger is moderated by the type of incentive. There is evidence that hunger increases dishonesty only when this directly helps alleviating the food-deprived state; that is, hungry participants showed increased cheating to get a snack pack, but not to get monetary rewards[42] (see also Orquin & Kurzban[43]). Other studies[15,44], however, found that scarcity in one domain (food) motivates maintaining resources in another domain (money). To illuminate this issue, we varied the type of incentive (food vs. money) in the DG in Study 4 (see Table 1 for an overview of the designs used in Studies 1–4).

In a more exploratory manner, in Study 4, we also investigated whether participants' state self-control mediates a potential relationship between hunger and selfishness. It has been suggested that the depletion of a resource (e.g., food) lowers self-control capacities, which, in turn, affects other acts that involve self-control[45,46]. Other studies support the idea that self-control is needed to engage in acts of prosocial behavior[47,48].

### Results Study 3 and 4

**Study 3 social value orientation.** In a simple linear regression, we found that subjective hunger did not significantly predict participants' SVO, $\beta = -0.10$, $t(265) = -1.58$, $p = 0.116$, $R^2 = 0.01$. When entering gender, age and field of studies into the regression to control for potentially confounding effects, results remained stable.

**Study 4 subjective hunger.** Participants before lunch reported significantly higher subjective hunger ($M = 7.74$, $SD = 1.94$) than participants after lunch ($M = 1.84$, $SD = 1.63$), $t(361) = 31.48$, $p < 0.001$, $d = 3.30$.

**Study 4 social value orientation.** We found no significant difference in SVO between participants before ($M = 21.51$, $SD = 10.41$) and after lunch ($M = 21.95$, $SD = 10.22$), $t(361) = -0.41$, $p = 0.686$, $d = 0.04$.

**Study 4 volunteering task.** We first analyzed whether the percentage of participants willing to volunteer in a future study differed between the two conditions, which was not the case (before: 45%, vs. after lunch: 47%), $\chi^2(1, N = 363) = 0.15$, $p = 0.696$, $\varphi = 0.02$. We further looked at the time (in minutes) that participants were willing to spend (among those who indicated willingness) as a continuous measure, and again found no difference (before lunch: $M = 45.00$, $SD = 59.14$, vs. after lunch: $M = 39.94$, $SD = 55.20$), $t(164) = 0.57$, $p = 0.569$, $d = -0.09$.

**Study 4 dictator game.** We conducted a 2 (time of measurement: before vs. after lunch) × 2 (type of reward: money vs. food) between-participants ANOVA on the amount of units (i.e., €0.60-units of an endowment of €6.00 in the monetary condition, or packs of nuts of an endowment of 10 in the food condition) that participants allocated to the other person. Our analysis revealed a significant main effect for type of reward, $F(1, 339) = 17.11$, $p < 0.001$, $\eta_p^2 = 0.05$. Participants were more generous when dividing food between themselves and another person (they allocated on average 5.29 out of 10 packs of nuts to the other person; $SD = 1.94$) than when dividing money ($M = 4.39$ out of 10 €0.60-units,

SD = 2.12; this translates to an average allocation of €2.63). More importantly, however, there was neither a main effect for time of measurement (before vs. after lunch), $p = 0.472$ (before lunch: $M = 4.78$. SD = 2.01; after lunch: $M = 4.93$. SD = 2.14), nor an interaction between time of measurement and type of reward, $p = 0.850$ (food condition: $M_{before\_lunch} = 5.19$, $SD_{before\_lunch} = 1.85$, $M_{after\_lunch} = 5.39$, $SD_{after\_lunch} = 2.03$; money condition: $M_{before\_lunch} = 4.33$, $SD_{before\_lunch} = 2.10$, $M_{after\_lunch} = 4.44$, $SD_{after\_lunch} = 2.16$).

**Study 4 state self-control**. Participants before lunch reported significantly lower self-control ($M = 4.86$, SD = 1.01) than participants after lunch ($M = 5.09$, SD = 0.93), $t(361) = -2.25$, $p = 0.025$, $d = 0.24$. We further tested whether self-control mediates the effect of lunch on the dependent measures using a bootstrapping approach (5000 samples) in the PROCESS macro for SPSS[49]. We found no significant indirect effects for any of the outcome measures.

## Complementary analyses
We conducted complementary correlational, meta-analytical, and Bayesian analyses to provide further specific tests of the effects of hunger on prosociality.

**Correlations with blood glucose levels**. We examined whether blood glucose levels in Studies 1 and 2 were correlated with the measures of prosociality. In both experiments, blood glucose levels at $t2$ (right before participants started with the tasks) were not significantly correlated to participants' prosociality in any of the measures that we included (Study 1: all $ps > 0.40$, Study 2: all $ps > 0.14$; within the two conditions and in the whole sample).

**Bayes factors**. To further examine whether our findings are generally in favor of the H0, that is, that hunger has no effect on prosociality, Bayes factors were calculated[50], using JASP[51]. We used a default prior (Cauchy scale: 0.707). Table 2 provides the Bayes factors of all dependent variables of all studies. The Bayes factors (BF01) indicate the relative likelihood that the data is in favor of H0 (rather than showing an effect of hunger). We consider a BF01 > 3 as at least moderate evidence in support of H0. We see in Table 2 that 7 of the 12 tests provide evidence for H0.

**Integrated effect sizes**. As direction and size of effects differed between the different outcomes (see Fig. 1 for effect sizes and confidence intervals of standardized effect sizes of all four studies), we conducted three meta-analyses to calculate the integrated effects[52] for (1) all dependent variables, (2) all dependent variables from the non-interdependent tasks, and (3) all dependent variables from the interdependent tasks. Positive effect sizes in Fig. 1 indicate effects in favor of the prediction of decreased prosociality when hungry. Negative effect sizes indicate increased prosociality when hungry. The integrated effect size including all outcomes of the four studies was $d = 0.073$, 95% CI [−0.024; 0.169]. The integrated effect size including only outcomes of the non-interdependent tasks (SVO, DG, social mindfulness, volunteering) was $d = 0.108$, 95% CI [0.001; 0.216], and for outcomes of the interdependent tasks (SHG, UG, PGG), it was $d = -0.076$, 95% CI [−0.296; 0.143].

## Study 5 lay theories about social effects of hunger
In four empirical studies, ranging from controlled randomized experimental approaches in the laboratory to correlational and quasi-experimental approaches in naturalistic settings, we found only very weak evidence that hunger reduces prosociality (and only in the integrated analysis for non-interdependent settings).

**Table 2 Bayes Factors (BF01). BF01 > 1 indicates that a null effect H0 is more likely than an effect of hunger on the given measure (with higher values indicating increasing likelihood)**

| Measure | Study | BF01 |
|---|---|---|
| PGG_Study1 | 1 | 8.48 |
| PGG_Study2 | 2 | 1.78 |
| PGG_ merged | 1 & 2 | 8.01 |
| SHG_Study1 | 1 | 5.98 |
| UG_proposer_Study2 | 2 | 1.84 |
| Social Mindfulness_Study2 | 2 | 2.33 |
| SVO_Study2 | 2 | 0.83 |
| SVO_Study3 | 3 | 1.73 |
| SVO_Study4 | 4 | 7.97 |
| DG_main effect hunger_Study4 | 4 | 6.85 |
| DG_hunger x resource IE_Study4 | 4 | 6.33 |
| Volunteering task_Study4 | 4 | 5.13 |

BF01 < 1 indicates that an effect of hunger is more likely than H0

These findings are not consistent with prior theorizing[1–6] and tentative empirical evidence[13–15]. Still, it is possible that the idea that acute hunger generally makes people more selfish is quite compelling, and perhaps even commonly held in society. To test whether our findings are indeed in contrast to public belief, we conducted a fifth study examining beliefs laypeople hold about social effects of hunger.

In an online survey, we described the DG of Study 4—in which hunger had no significant effect—to participants ($N = 197$) of different demographic backgrounds, provided the actual mean amount of money that was shared in the control (after lunch) condition (i.e., €2.67) in Study 4, and then asked them to estimate how much was shared in the hungry (before lunch) condition. The correctness of guesses was incentivized. We further asked participants about their beliefs regarding people's general behavior when acutely hungry and their own behavior when acutely hungry.

## Study 5 results
In the DG in Study 4, we found that participants in the hungry (before lunch) condition shared on average €2.60, while participants in the control (after lunch) condition shared on average €2.67. In Study 5, we provided participants with the mean amount shared in the control condition as a reference point (i.e., €2.67) and asked them to estimate the amount for the hungry group. We used a one-sample $t$ test to assess whether their mean estimate of the amount of money shared in the hungry condition was significantly different from the provided value of 2.67 (control condition). Participants on average estimated that the hungry group shared €1.96 ($SD = 1.13$), which is significantly less than €2.67, $t(196) = -8.83$, $p < 0.001$, $d = 0.63$. A second one-sample $t$-test revealed that their mean estimate for the hungry group ($M = 1.96$) also differed significantly from our actual result for the hungry group in Study 4 (€2.60), $t(196) = -7.97$, $p < 0.001$, $d = 0.57$.

Also, a clear majority of participants held beliefs that hunger undermines prosociality. Most participants indicated that they believed hungry people to be more selfish (79%), less cooperative (73%), and less helpful towards others (68%). They believed the same to be true for themselves (62% more selfish, 68% less cooperative, and 59% less helpful towards others).

## Discussion
Does acute hunger undermine prosociality, as prior theorizing[1–6] and prior empirical evidence[13–15] would suggest? We found that

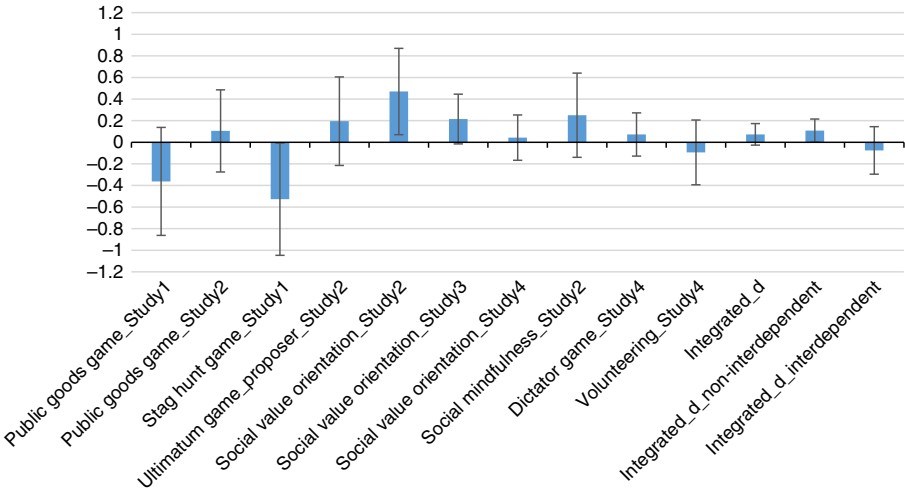

**Fig. 1** Effect sizes *d* of the effects of hunger on prosociality. Positive values indicate effects in favor of a hypothesis that hunger decreases prosociality. Negative values indicate increased prosociality when hungry. Non-interdependent tasks: SVO, Social Mindfulness, DG, Volunteering. Interdependent tasks: PGG, SHG, UG. Effect size calculation was based on Lipsey and Wilson's formulas[52]. Error bars indicate 95% confidence intervals

laypeople clearly predict individuals to share less with others when they are hungry (Study 5). However, our empirical results do not provide support for this general belief. Using a variety of methods, in both non-interdependent and interdependent settings, we found no differences between people in a hungry and a non-hungry state in their levels of prosociality, as indicated by significance tests and Bayes factors. These studies, two of which were pre-registered, involved experimental inductions of hunger in the laboratory as well as self-reported hunger in the field. The integrated effect size including all DVs from our four studies was close to zero ($d = 0.076$) and the 95% CI included zero [−0.024; 0.169]. The same was true for the integrated effect size for interdependent tasks ($d = -0.073$, CI [−0.296; 0.143]). There was a very small integrated effect size of $d = 0.108$, 95% CI [0.001; 0.216] for non-interdependent tasks. Hence, the overall picture does not point to a general or pronounced detrimental effect of hunger on prosociality. Only for the integrated non-interdependent tasks there was, however weak, evidence for the prediction that hunger decreases prosociality.

How can we explain this discrepancy between past publications and common belief on the one hand and our findings on the other hand? A tentative explanation could be that beliefs about hunger and prosociality are deeply grounded in the widespread "myth of self-interest"[53], a strong belief that people are ultimately selfish[54]. As part of this belief, people may also think that individuals fall back on self-interest when stakes are high. Hence, they may overestimate the extent to which self-interest rather than prosocial motives are triggered, when resources are limited. Moreover, people might underestimate the social context that dictates cooperation in some cases, even if self-interest was triggered. This might be particularly true for situations that represent interdependence, as in such situations hunger could not directly translate into selfishness. Our analyses revealed a—very small— effect of hunger on prosociality in non-interdependent contexts; that is, when the outcome an individual receives depends exclusively on her decisions. These are prototypical situations in which giving in to selfish impulses is not believed to result in repercussions—such as rejection decisions in the UG or increased selfishness by others in interdependence tasks. Strikingly, even in such situations, which in our case were also stripped of elements of social resonance or non-anonymity, we found only a very small negative effect of hunger.

We suggest that most conflicts between self-interest and prosociality in everyday life entail some degree of interdependence— a feature captured by the interdependent tasks used in our studies. For example, in an UG, participants cannot simply reduce their contributions to protect their resources without increasing the danger of eliciting a negative response by others that leads to receiving no rewards at all, when the offer is rejected. The null findings in these situations might indicate that, even if hunger increases the urge to acquire more resources, people are still able to consider their interdependence with others and the strategic constraints of the situation, which is in line with evidence that hunger even improves strategic decision making[55]. In non-interdependent situations, people might at least be aware of social norms and might be guided by them in their behavior[56], as the non-significant findings and the only small integrated effect size for these tasks indicate. More generally, our findings are in line with the argument that people are able to compensate for their psycho-physiological impairments when they perceive it as required by the social context[23,57].

The interpretation that hunger increases egoistic impulses that are suppressed by social requirements is in line with counter-vailing direct and indirect effects found in Aarøe and Peterson's[17] study. It is also in line with our finding in Study 4 that hunger significantly reduces perceived self-control, which does not translate into more selfish behavior. People might perceive themselves to be less able to control impulses when hungry but, in the context of social interaction, there is a range of other countervailing impulses at play, such as fairness or risk of rejection. The experience of lowered self-control when hungry might even be a foundation of the lay theories that hunger decreases prosociality, as shown by study 5. Speculatively, people might be able to accurately judge the impact that acute hunger has on their self-control, but unable to correctly anticipate the influence of social factors that maintain prosociality.

We should also note the possibility of hunger effects being overestimated in published literature. Strikingly, in Study 4, we found that hunger exerts no effect on selfishness even when the resource at question is useful to directly eliminate acute hunger, that is, food rather than money. Looking at the psycho-biological level, a recent meta-analysis[43] suggests that the spectrum and intensity of behavioral and psychological consequences of fluctuations in blood glucose levels have been overestimated in

previous research. In line with this, in our studies, we also found no significant correlations between blood glucose levels and prosociality.

To avoid misunderstandings, note that we examined hunger in terms of an acute condition within the upper boundaries of the natural daily fluctuation typical in Western industrialized societies (Studies 1 to 4 were conducted with undergraduate student samples in Germany). The effects of hunger in terms of a chronic, potential life-threatening state, as a consequence of food scarcity due to poverty, natural disasters, or war is beyond the scope of the present paper. Rather, our primary aim was to replicate and extend previous research and theorizing on the effects of acute hunger and/or blood glucose on prosociality[13–17]. All of these studies used manipulations similar to ours or weaker, for example, the application of a carbonated versus non-carbonated soft drink without prior food deprivation[14,17]. Most of these studies reported an effect of hunger or blood glucose on prosociality. Hence, in terms of reproducibility and generalizability, as well as in terms of advancement of theory development, we therefore deem our new evidence in support of the null hypothesis to be highly informative.

We are fully aware of the qualitative difference between acute hunger as a non-critical state and chronic hunger as an existential threat. Nevertheless, we believe that our manipulations assess non-trivial forms of hunger, as many people in Western industrialized worlds might experience them. For example, our variations in hunger were validated by strong and consistent effects on two key aspects of hunger: blood glucose levels (measured in Studies 1 and 2), and self-reported hunger (measured in Studies 1, 2, and 4). As natural fluctuations in blood glucose are regulated within a narrow range, even short food deprivation is likely to cause blood glucose levels to be at the lower boundary of the regulatory corridor[58,59]. Notably, we did not detect any significant link between levels of blood glucose and degree of prosociality in Studies 1 and 2.

Although it is beyond the scope of this paper, we believe that—given the strength of our manipulations and the small range of natural blood glucose levels—our findings provide a first indication that also severe forms of hunger might not per se undermine prosociality. From an evolutionary perspective, it could be argued (albeit speculatively) that shortage of food should even foster prosociality and cooperation. For example, Gurven[60] emphasized that in small scale societies meeting daily food needs is a highly interdependent task. Interestingly, in such societies scarce food resources even increase prosociality (peaceful sharing) rather than selfishness. This tendency might be hard-wired as a consequence of mankind's long history of cooperative hunting and gathering. Evidence from evolutionary game theory[61] supports the idea that in times of scarcity (i.e., rare pay-offs), evolution favors strategies that minimize the risk of not receiving any resources at all rather than strategies that maximize resources, which also relates to the idea that, when facing limited resources, people might strive to cooperate.

Taken together, the idea that hunger undermines prosociality seems a compelling belief, and there has been some evidence in support of it. However, given the null results that we found, we conclude that hunger is unlikely to substantially undermine prosociality. This seems to be especially true in social contexts characterized by some degree of interdependence. More generally, our results indicate that people seem to be able to overcome some state of deprivation and to detect and respond to the actual or perceived social requirements of the situation. In conclusion, we suggest that hunger often does not translate into more selfishness because many situations share some elements of interdependence − when other people notice our actions and are able to respond to them. Humans are social animals, after all.

## Methods

**Test-power calculation for Studies 1 and 2**. To avoid inflation of type-one errors, levels of significance were Bonferroni-corrected (Study 1: $\alpha = 0.017$; Study 2: $\alpha = 0.008$). Test-power calculations (G*Power 3.1.9.2)[62] showed that sample sizes of both studies were sufficient to detect medium to large effects with the adjusted alphas and power = 0.85. We aimed at examining effects with at least medium size based on effect sizes (and sample sizes) of earlier research as well as on considerations regarding practical relevance. Of the earlier papers that reported the predicted effect of hunger on prosociality, only one[17] reported effect sizes (or provided the data required for computing effects sizes). In the first of two studies resembling our design (Study 1: food deprivation prior to participation in the control condition before measurement of prosociality; though food deprivation was only for 4 h) the authors found a moderate effect. In the second study, which used a weaker manipulation of hunger (Study 2: olfactory food cues in saturated participants), a small to moderate effect was found. The sample sizes of the earlier studies ranged from $N = 58$ to $N = \sim130$ (one study[13] did not report the Ns for the relevant conditions). With respect to practical relevance, we used strong manipulations of hunger and blood glucose levels, that were also stronger as compared to the earlier studies that reported the predicted effect. Hence, our manipulations should produce stronger variations in hunger and blood glucose compared to common fluctuations occurring in industrialized societies. We, therefore, argue that the commonly occurring fluctuations would be of relevance only if our strong (but still ecologically valid) manipulations produced at least medium effects. Further, our experimental studies were complemented with correlational and quasi-experimental studies that were clearly higher powered as compared to earlier studies ($N = 267$ and 363, respectively).

**Study 1 participants and procedure**. Sixty-two undergraduate students (81% female, mean age = 22.61) were randomly assigned to a one-factorial between-participants design (hunger vs. control). Experimental sessions were scheduled at 10 a.m. All participants were instructed not to eat anything after 10 pm the previous night. Baseline measures ($t1$) of subjective hunger (one-item visual analog scale: "How hungry are you right now?"; ranging from 0 to 10) and blood glucose levels (Medisana® MediTouch) were obtained. Next, participants in the control condition took part in a "tasting experiment" and ate two chocolate puddings (total sugar 36.75 grams). After a latency of 10 min, subjective hunger and blood glucose levels were measured again ($t2$), and participants started the tasks. Participants' payment was dependent on individual payoffs from the tasks ($M = €21.16$, SD = 3.01). Study 1 was approved by the local ethics committee of the University of Hildesheim, Germany. Informed consent was obtained, and participants were fully debriefed.

**Study 1 public goods game**. Participants played a one-shot computer-mediated PGG with two other anonymous participants. Participants received €8 and decided how much money they wanted to keep for themselves and how much they wanted to donate to a common pool (in increments of €1). All money donated to the common pool was multiplied by 1.5 and afterwards split equally among the three participants. Each participant received the money kept for herself plus one third of the common pool.

**Study 1 stag hunt game**. Participants played a one-shot SHG with another anonymous participant. Participants had to decide between a cooperative, high pay-off, but socially risky option ('hunting a stag') and an uncooperative, low-pay-off, but safe option ('hunting a hare'). Hunting the stag resulted in a higher pay-off (€2.5), but only if both players simultaneously decided to hunt the stag. The hare, in contrast, could be bagged independent of the decisions of the other player, but resulted in a lower payoff (€1).

**Study 1 exploratory measures**. Study 1 also contained a non-social risk task. Hunger did not affect non-social risk.

**Study 2 participants and procedure**. In Study 2, we increased both the strength of the experimental manipulation and the sample size: first, experimental sessions were scheduled at 12 p.m. instead of 10 a.m. Second, in the control condition, participants consumed a higher amount of sugar (42 g) with a faster uptake into the blood stream (grape juice, glucose, and buns). Third, the latency between food consumption and experimental tasks was increased to 15 min. One-hundred-three students (67% female, mean age = 24.06) were randomly assigned to a one-factorial between-participants design (hunger vs. control). Participants' payment was dependent on individual payoffs from the tasks ($M = €24.43$, SD = 2.75). Study 2 was approved by the local ethics committee of the University of Hildesheim, Germany. Informed consent was obtained, and participants were fully debriefed.

**Study 2 public goods game**. We used the same procedure as in Study 1, with the exception that participants received €10 in Experiment 2 to allow for greater variance.

**Study 2 ultimatum game**. Participants played a UG[63], which involves two players, a "proposer" and a "responder". First, participants acted in the role of the proposer.

They received €10 and were asked to propose a division between themselves and the responder (another participant who would allegedly participate in the study later). Participants were told that the responder would decide whether to accept or reject the offer. If the offer was accepted, both the proposer and the responder would be paid accordingly. If the offer was rejected, both would receive a payoff of zero. In fact, all participants received a payment according to their offer at the end of the study, as no future participants decided on these offers. For exploratory purposes, our participants also played the UG in the responder role. Note that it could be argued that hunger should increase or decrease the acceptance of unfair offers. Hunger had no effect.

**Study 2 social value orientation**. We measured SVO using the 6-items version of the SVO slider measure[25]. Participants indicated their preferences for hypothetical distributions of money between themselves and another person in a set of non-constant-sum dictator games. High values in the slider measure indicate a more prosocial orientation. As SVO can be conceptualized both as a trait and a state[8,25], participants filled in the SVO measure twice: during the experiment to determine the influence of hunger on SVO, and 1 week before the experiment to determine their baseline social preferences.

**Study 2 social mindfulness**. Participants completed the social mindfulness task[27]: in a series of computer-based trials, they chose between three objects of one category (e.g., three pens, three wrapped gifts). Two of these were always identical; the third differed in one aspect (e.g., color). Participants were asked which object they would choose if another person was to choose after them and the object chosen was not to be replaced. A choice of one of the two redundant objects was coded as the socially mindful choice, as it left the hypothetical other person in control about which object they would get.

**Study 2 exploratory measures**. Study 2 also contained exploratory measures of self-reported trust, personality, and moral decision making. Hunger did not affect any of these measures.

**Study 3 participants and procedure**. We conducted a power analysis (using G*Power 3.1.9.2[62]) given a potential participant pool of $N = 300$, and taking an alpha of 0.05 and power of 0.80, for a linear regression model with one predictor. We would be able to detect a small effect of $f^2 = 0.02$.

Two hundred and seventy-six students were recruited in four different lectures at a German University and were asked to fill in a short paper-and-pencil survey. Nine surveys were returned incomplete, so 267 participants (79% female, mean age = 22.11) were included in the analysis. Prior to data collection, Study 3 was pre-registered on the OSF website (https://osf.io/qxa6t/?view_only=339cbd05be6b44b0b03afed1e09c2c10) and approved by the local ethics committee of the Justus-Liebig-University Gießen, Germany. Informed consent was obtained, and participants were fully debriefed.

The survey consisted of demographics, subjective hunger, and SVO in counterbalanced order to control for possible order effects. Furthermore, we investigated relationships with tiredness as a separate research question, which is not subject of discussion here. Participants did not receive any financial rewards.

**Study 3 subjective hunger**. Participants were asked how hungry they felt at this moment and indicated their response on a scale from 1 (not at all) to 10 (very hungry). Additionally, participants indicated the time of their last meal. We used time since the last meal as a second indicator of hunger, which revealed the same results as our analysis with subjective hunger.

**Study 3 social value orientation**. We used the six primary items of the SVO slider[25].

**Study 4 participants and procedure**. We recruited 363 participants (62% female, mean age = 23.57) in front of the cafeteria at a German University at lunchtime. A sensitivity analysis (using G*Power 3.1.9.2[62]) revealed that, given our sample size of $N = 363$, alpha of 0.017 (Bonferroni-corrected), and power of 0.95, we could detect a small to medium size effect of $f = 0.21$ (main effect for hunger).

Participants were on their way to lunch (i.e., entering the cafeteria, $n = 179$) or returning from lunch (i.e., exiting the cafeteria, $n = 184$). They were randomly assigned to either a monetary reward condition or a food reward condition; hence, we used a 2 (before vs. after lunch) × 2 (monetary vs. food reward) between-subjects design with time of measurement (before vs. after lunch) as a quasi-experimental factor. Prior to data collection, Study 4 was preregistered on the OSF website (https://osf.io/8n7a9/?view_only=42fcd0af37ca4891a30470e36c729278) and approved by the local ethics committee of the Justus-Liebig-University Gießen, Germany. Informed consent was obtained, and participants were fully debriefed.

The survey included demographics, three different measures of prosociality, and a measure of state self-control. After completing the survey, participants received their reward based on their decision in the DG (i.e., either money or food, depending on experimental condition). As in Study 3, we investigated relationships with tiredness as a separate research question, which is not subject of discussion here.

**Study 4 subjective hunger**. Participants indicated their momentary hunger on a scale from 1 (not at all hungry) to 10 (very hungry).

**Study 4 social value orientation**. We used the six primary items of the SVO slider[25].

**Study 4 volunteering task**. We adopted a volunteering task from McClintock and Allison[28] assessing participants' willingness to participate in future psychology studies without obtaining financial reward. If participants agreed, they had to indicate the number of minutes they would be willing to contribute (in 15-min steps up to 300 min).

**Study 4 dictator game**. Depending on the experimental condition, participants either received €6.00 (monetary reward) or ten 40 g packs of nuts and dried fruit (food reward) that they could divide between themselves and an anonymous future participant. They could allocate money or food to the other participant (i.e., €0 – €6.00 in steps of €0.60, or 0 – 10 packs of food). They were informed that they would keep the remaining amount, that they would not meet the other participant and that their decision would remain anonymous. The donations made in the DG were paid out to subsequent participants, after their participation. Prior to participation, participants were not informed that they would receive additional money at the end of the experiment. Moreover, participants were not informed that the future participants would also be in the role of a dictator.

In order to select an attractive reward for the food condition, we conducted an online pre-test including 216 participants who rated five different snacks. Nuts and dried fruit were perceived as most valuable among the five snacks (mean estimate = €0.67 per pack) and were the first choice of most participants (36.6%) when sorting the snacks according to their preference. In our main study, participants further indicated how much they liked nuts and dried fruit on a scale from 0 (not at all) to 4 (very much), to control for bias due to participants being generous only because of their food preferences. For the analysis regarding the DG, we excluded 20 participants in the food reward condition, who indicated that they did not like nuts and dried fruit at all (response 0 on the 0–4 scale).

**Study 4 state self-control**. We used the short version of the State Self-Control Capacity Scale in German (SSCCS-K-D)[64], containing 10 items (e.g., "I feel awake and concentrated", scale 1 = not at all true to 7 = entirely true; 8 items reversed). High values indicate high self-control. Cronbach's $\alpha$ was 0.85.

**Study 5 participants and procedure**. We conducted an online study via Amazon MTurk with $N = 210$ participants. Thirteen participants were excluded from the analyses as they failed to answer the attention check correctly, resulting in a sample of $N = 197$ (mean age = 37.36, 37% female). Adopting a procedure from Lee, Frederick, and Ariely[65], participants read an accurate description of our Study 4 and were informed about the mean amount of money that was shared in the DG in the after lunch (control) condition (i.e., €2.67). We then asked participants to estimate the mean amount of money that was shared in the before lunch (hungry) condition (given possible values between €0.00 and €6.00). They received an additional bonus of $0.50 for accurate estimates (i.e., within a 20% range around the actual result for the hungry group).

Subjects were then asked three questions on how they thought people in general would behave when hungry ("Do you think people are more or less selfish/cooperative/helpful when they are hungry?"), and afterwards, how they thought they themselves would behave when hungry ("Do you think you are more or less selfish/cooperative/helpful when you are hungry?"). Each question had three possible choices (i.e., "more selfish/cooperative/helpful", "no difference", "less selfish/cooperative/helpful"). Participants were paid according to US American minimum wage. Prior to data collection, Study 5 was pre-registered on the OSF website (https://osf.io/3cvnz/?view_only=aadb4be8238844d68867180057ec91f8), reviewed and approved by the University of Oxford's Central University Research Ethics Committee, with the reference number MS-IDREC-R56657/RE001. Informed consent prior to participation was obtained.

## Data availability

The data that support the findings of this paper are available on the OSF website (https://osf.io/zexd7/?view_only=480593713c904397a033e751a6da7a69).

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

## Acknowledgements
This research was supported by a grant from the Volkswagen Foundation awarded to J.A.H.
and N.S.F.

## Author contributions
All authors planned the experiments. J.A.H., C.S., N.S.F., A.M. and J.L. prepared
the study materials and conducted the studies. J.A.H., C.S., and N.S.F. analyzed the data.
J.A.H., C.S., N.S.F. and P.V.L. wrote the manuscript. All authors provided feedback at
different stages of the research, reviewed and approved the manuscript.

## Competing interests
The authors declare no competing interests.

## Additional information
019-12579-7.

**Peer review information** *Nature Communications* thanks Michael Petersen and the
other, anonymous, reviewers for their contribution to the peer review of this work. Peer
reviewer reports are available.

**Publisher's note** Springer Nature remains neutral with regard to jurisdictional claims in
published maps and institutional affiliations.

