## [Peer Review File · Nature Communications]

Reviewers' comments:

Reviewer #1 (Remarks to the Author):

This is a review of "Does Hunger Undermine Prosociality and Cooperation?". This manuscript examines whether hunger influences prosociality and cooperation as measured by both self-reported measures and economic game behavior. Overall, the authors conclude that the presented evidence suggests that effects are most likely zero. To this null finding, the authors add the interesting twist of documenting that, despite these results, lay people have a strong intuition that hunger increases self-interested behavior. To support their claims, the authors reports four studies that test the key claim and two of these are pre-registered.

1) On the bottom of page 3 the authors write: "Another study¹⁶ found a marginally significant effect of experimentally induced hunger on the amount of money participants shared in a dictator game (DG)." This is not an entirely accurate description of the study in question. The study found a marginally significant direct effect of observed (but experimentally influenced) blood glucose levels on DG behavior but this was qualified by a counter-veiling indirect effect such that the combined effect was null. Thus, this particular result of the study is entirely in line with the present findings. As such, the statement on page 13 is also inaccurate: "What we found in four studies (Studies 1-4) is in contrast to these beliefs and prior empirical evidence^{13, 16}." The relationship with prior findings should thus be more clearly and accurately communicated.

2) I believe that Study 3 is potentially at risk of suffering from omitted variable bias. Presently, it is a bivariate correlation between subjectively experienced hunger and social value orientations. But can it really be true that there are no potentially confounding variables? This ought to be discussed and conclusions should be calibrated to the lack of control for relevant variables. Furthermore, it would strengthen the conclusions, if the authors conducted exploratory analyses with the potential confounds that they did in fact obtain measures of (e.g., sex).

3) I would encourage the authors to discuss their self-reported measures more (e.g., social value orientations). What are the constructs these measures are designed to capture and how does these constructs relate to the research question? And what are the expectations for these measures? As the authors write, there is a common-sense view that hunger increases self-interested behavior. At the same time, as the authors also discuss, such self-interested reactions could increase support for collective sharing (e.g., social welfare). Given that the self-reported measures are 'cheap talk' that could be used for such strategic morality, it is not entirely clear to me what the authors expect and why. This is also complicated by (1) the fact that there is some evidence that hunger increases social value orientations – and, in fact, makes people more prosocial - and (2) the discussion of the anthropological literature on p. 14, which points in the same direction. Thus, I think both the theoretical expectations, the direction of the empirical effects and the weight of the overall evidence requires more discussion.

4) It is unclear from the manuscript how the power analysis for Study 4 is conducted. The power analysis refers to a "conventionally medium size effect of $f = 0.25$ " but what exact effect is this expected to reflect? The experimental is a factorial 2×2 design. Is it the interaction between the experimental factors, which is expected to equal .25, or is the main effect of hunger (i.e., averaging across the food vs. money conditions)? If it is the interaction, does the power analysis then account for the fact that an interaction is harder to detect than a main effect? (due to the higher standard errors that necessarily occurs because of the multicollinearity between interaction terms and the main terms).

In addition to these major concerns, I have the following minor comments:

1) Study 4 reveals a significant effect of hunger on self-control. To me this seems to provide an obvious explanation for the lay theory that hunger increases self-interest. People (correctly) feel less able to control impulses when hungry and self-interest is certainly a strong impulse. What they might be forgetting is that, in the context of social interaction, there are a range of other countervailing impulses at play such as fairness or reputation. From an adaptationist perspective,

such factors are just as 'hard-wired' (and, hence, impulse-based) as pure self-interest. If the authors agree, the finding related to self-control might be worthwhile to highlight in the discussion.

2) To help the reader assessment the strength of the underlying evidence, I think it would be helpful to have the Ns of each individual study when the results of the studies are described and also provide more elaboration on the nature of experimental manipulations. Everything is in the methods section but, presently, it is difficult to assess the results without going back and forth between the results and the methods section.

3) The authors write: "The data that support the findings of this study are available from the corresponding author upon reasonable request." Given the authors' laudable commitment to open science (as reflected in the use of pre-registration), I was curious why they wouldn't also just share the data and analyses files in a public repository?

Reviewer #2 (Remarks to the Author):

This paper investigates whether hunger affects prosocial behavior. Several experimental and non-experimental methods are used to create variation in hunger. Several instruments are used to assess prosociality. Overall, no effect is found. This implies that the relevance of the study has to be assessed based on the relevance of the research question. I consider it as relevant because there is empirical evidence for the existence of a hunger effect. In addition, the authors show that lay people believe that hunger reduces prosociality.

General comments

The hunger induction is well done. In Study 1 and 2 participants are recruited in a hungry state and half of them get food; in Study 3 a correlational approach based on subjective hunger is used; and in Study 4 students are recruited before and after lunch. It is also positive that Study 3 and 4 are rather high powered and preregistered.

The measurement of prosociality is based on several games. However, only social value orientation and the dictator game provide undistorted measures of prosociality. In the other games, the behavior of the player does not only depend on the prosociality but also on how the participants assess the behavior of the other players. This is obvious in the ultimatum game, the trust game and the stag hunt game where also selfish players behave prosocially if they believe in the prosociality of the other player(s). Vice versa, in the public goods game, also prosocial (conditionally cooperative) players behave as selfish players if they believe that the other players are selfish. This makes the analysis difficult. It is not so clear that the different measures measure the same concepts. So it is not clear how the evidence should be aggregated. The authors use a Bonferroni correction in order to take multiple comparisons into account. This is safe not to overestimate significance, but it is not the best argument for a null result. Actually, the integrated measure that considers only the SVO, which is truly prosocial behavior, shows a significant effect in the expected direction. I wonder whether a principal component analysis or a method using latent concepts could replicate the "established" result.

In the ultimatum games deception was used. It is known that deception can reduce participants' credibility in the experimenter. However, I am aware of the fact that is not considered as a problem in psychology. In the dictator game the participants could donate to an anonymous future participant. It is not clear how this is done. It is important to know whether the recipients were also in the role of the dictator because in this case no transfer by all dictators is also fair.

It is fine to separate methods from the main text. However the main text should contain the all relevant information. For example, it is admittedly natural to induce hunger by giving food to the control group. Explicitly mention this in the text would help the reader to immediately understand that the treatment group (hunger) is the untreated group.

Assessment

The paper addresses a relevant research question with a comprehensive study. It produces a null result but this null result will not be undisputed. I recommend inviting a revision, which improves clarity in the main text, provides a better theoretical foundation of the different measures, and analyses the data more deeply.

Specific comments

1. The paper should be clearer in the discussion of the measures of prosociality (see above). For example, consider economic papers that game theoretically analyze these games assuming non-selfish preferences as Ernst Fehr and Klaus M. Schmidt. 1999. "A Theory of Fairness, Competition, and Cooperation." *Quarterly Journal of Economics*, 114 3, 817-68. Provide also reference to the original papers of the different game (UG, trust game...).
2. The analysis should take the nature of the different games into account (see above).
3. Page 3, „Keeping versus using resources is particularly relevant for cooperation, as cooperating with others requires investing one’s own resources (e.g., money, time, effort) to achieve collective goals.“ Keeping versus using resources is relevant in all dimensions of prosocial behavior. Actually, most definitions of prosocial behavior assume that prosocial requires giving up some resources in order help others at a cost.
4. Page 5. Trust is considered as non-interdependent measure of prosociality. I disagree with this view. Trust is highly interdependent. It relies on the others’ trustworthiness. So it depends on the expected trustworthiness of the trustee.
5. Page 6. In the results the order of the treatments varies across the different measure. This makes it more difficult to read.
6. Page 10, presentation of the dictator game. Report the all four transfers of the 2x2 design and check separately whether hunger affects decisions in the money and food treatment. Alternatively use a regression.
7. In the lay study, the correctness of the guess was not incentivized. It could cause the participants to respond more in line with an experimenter demand effect.

Typos

8. page 5. “non-interdepended”
9. Check again the references. The second author in reference 19 is Kurzban and not Kurzbahn.

Reviewer #3 (Remarks to the Author):

This paper reports results from a number of studies testing whether hunger reduces people’s prosocial and cooperative tendencies. The hypothesis being tested is that a scarcity of resources causes individuals to prioritize their self-interest, becoming less prosocial and less cooperative. The authors conduct a series of studies testing their hypothesis. They put their subjects through a series of games and surveys either in a hungry or not hungry state. They report no evidence that hunger results in less prosocial or cooperative behavior.

The hypothesis is well worth testing, but I do not believe these studies are able to really address the issue. The primary problem is hunger in the context of these studies is just a minor inconvenience. Having not eaten since 10pm the previous evening is unlikely to be a major issue for most subjects, especially given they know that they can feed in a short period of time. This is particularly true for those just about to enter the cafeteria. This form of hunger is nothing like the hunger someone experiences if they are suffering deprivation due to a true food shortage brought on by a drought or other catastrophic events. I believe the null result is because the treatment is just too weak to have any significant effect on behavior. I do not think we have learned anything.

Minor issues:

1. For the non-social risk task, just counting the number of risking choices can mask problems with this type of risk attitude measure. It is not uncommon in tasks such as this for subjects to make inconsistent choices. The authors do not address this issue. Do their subjects make inconsistent choices?
2. The ultimatum game is not a good measure of prosocial attitude. Keeping more for yourself is a risky decision; the probability of a rejection increases with the amount kept. So a more generous offer may reflect risk aversion rather than prosociality.

Reviewers' comments

Reviewer #1

Overall, the authors conclude that the presented evidence suggests that effects are most likely zero. To this null finding, the authors add the interesting twist of documenting that, despite these results, lay people have a strong intuition that hunger increases self-interested behaviour. To support their claims, the authors reports four studies that test the key claim and two of these are pre-registered.

We are delighted to hear that you found our paper interesting.

1) On the bottom of page 3 the authors write: "Another study found a marginally significant effect of experimentally induced hunger on the amount of money participants shared in a dictator game (DG)." This is not an entirely accurate description of the study in question. The study found a marginally significant direct effect of observed (but experimentally influenced) blood glucose levels on DG behavior but this was qualified by a counter-veiling indirect effect such that the combined effect was null. Thus, this particular result of the study is entirely in line with the present findings. As such, the statement on page 13 is also inaccurate: "What we found in four studies (Studies 1-4) is in contrast to these beliefs and prior empirical evidence." The relationship with prior findings should thus be more clearly and accurately communicated.

Thank you very much for bringing this issue to our attention. Indeed, we agree that our presentation of the study by Aarøe & Petersen (2013) was somewhat oversimplified. Following your suggestion, we now explicitly describe the countervailing direct and indirect effects and provide more details on the study (and also on the other experimental studies, particularly distinguishing between effects of subjective hunger and blood glucose), see p 4.:

"In one study¹⁵, food deprivation reduced participants' intention to donate money to charity, and in another study¹³, food abundance, induced by participants' consumption of an energy bar prior to their decisions, resulted in stronger intention to donate money to charity. In line with this, there is also tentative evidence that increased blood glucose levels lead to higher contributions in the DG¹⁴. The picture, however, is somewhat inconsistent, since two other studies found no significant effect of experimentally manipulated hunger on charitable giving¹⁷ or experimentally manipulated blood glucose levels on the amount of money participants shared in a DG¹⁶. A closer look at the latter null effect, however, revealed that it was due to two countervailing effects of blood glucose on prosociality¹⁶. A marginally significant direct effect of blood glucose on the amount of money shared in the DG is in line with the idea of resource accumulation and consistent with the previous findings. At the same time, blood glucose levels were also negatively related to support for social welfare, which, in turn, was positively related to sharing behavior. Hence, via this indirect effect, decreased blood glucose may have led to increased prosociality. This finding could be interpreted in terms of low blood glucose levels increasing selfish tendencies, but these are cancelled out by maintenance or even strengthening of prosocial norms."

Also, in the discussion, we further elaborate on the countervailing direct and indirect effects that are in line with the overall pattern of our results, see p. 19:

"The interpretation that hunger increases egoistic impulses that are suppressed by social requirements is in line with countervailing direct and indirect effects found in Aarøe and Peterson's¹⁶ study. It is also in line with our finding in Study 4 that hunger significantly reduces perceived self-control, which does not translate into more selfish behavior. People might perceive themselves to be less able to control impulses when hungry but, in the context of social interaction, there is a range of other countervailing impulses at play, such as fairness or risk of rejection. The experience of lowered self-control when hungry might even be a foundation of the lay theories that hunger decreases prosociality, as shown by study 5. Speculatively, people might be able to accurately judge the impact that hunger has on their self-control, but unable to correctly anticipate the influence of social factors that maintain prosociality."

2) I believe that Study 3 is potentially at risk of suffering from omitted variable bias. Presently, it is a bivariate correlation between subjectively experienced hunger and social value orientations. But can it really be true that there are no potentially confounding variables? This ought to be discussed and conclusions should be calibrated to the lack of control for relevant variables. Furthermore, it would strengthen the conclusions, if the authors conducted exploratory analyses with the potential confounds that they did in fact obtain measures of (e.g., sex).

Thank you for this interesting comment. Study 3 was designed as a first parsimonious study using natural fluctuation in hunger, also piloting Study 4. Hence, in this study we only measured SVO, subjective hunger, and some demographics (sex, age, field of studies). We had decided in advance that time and other constraints would preclude a longer questionnaire. Thus, we cannot fully rule out the possibility of omitted variable biases, but we conducted the exploratory analysis, as you suggested, and found no evidence of omitted variable biases. In the analysis of SVO, for exploratory purposes, we added sex, age, and participants' college major as predictors. The latter was dichotomized, as the sample consisted of psychology students ($n = 121$) and students of various other majors with a minor in psychology ($n = 153$; predominately teachers training in diverse fields, such as science, math, sports, languages). None of the three demographic variables was correlated with the predictor subjective hunger (all r s $< \pm .09$, all p s $> .15$, and only sex was correlated with the outcome SVO ($r = -.122$, $p = .046$; indicating higher prosociality in female as compared to male participants). When the demographics are entered in the regression together with subjective hunger, the coefficients for subjective hunger remain virtually identical (without demographics: $\beta = -.10$, $t(265) = -1.58$, $p = .116$, $R^2 = .01$; with demographics: $\beta = -.096$, $t(265) = -1.57$, $p = .119$, $R^2 = .01$).

We now provide this information in the result section of Study 3, see p. 12:

"In a simple linear regression, we found that subjective hunger did not significantly predict participants' SVO, $\beta = -.10$, $t(265) = -1.58$, $p = .116$, $R^2 = .01$. When entering gender, age and field of studies into the regression to control for potentially confounding effects, results remained stable."

3) I would encourage the authors to discuss their self-reported measures more (e.g., social value orientations). What are the constructs these measures are designed to capture and how does these constructs relate to the research question? And what are the expectations for these measures? As the authors write, there is a common-sense view that hunger increases self-interested behavior. At the same time, as the authors also discuss, such self-interested reactions could increase support for collective sharing (e.g., social welfare). Given that the self-reported measures are 'cheap talk' that could be used for such strategic morality, it is not entirely clear to me what the authors expect and why. This is also complicated by (1) the fact that there is some evidence that hunger increases social value orientations – and, in fact, makes people more prosocial - and (2) the discussion of the anthropological literature on p. 14, which points in the same direction. Thus, I think both the theoretical expectations, the direction of the empirical effects and the weight of the overall evidence requires more discussion.

Thank you for pointing out the lack of clarity with regard to the measures and their theoretical underpinnings. In the revised version of the manuscript, we rewrote the whole theory section to clarify the nature of the measures used in the studies, see p. 3-7 (see also Reviewer 2, comments 1 and 4, and Reviewer 3, comment 3):

"It has been argued that hunger, as a signal of limited resource availability, reduces prosociality, that is, the willingness to invest one's own resources (e.g., money, time, effort) to help others^{7, 8}. Such "pure prosociality" is typically measured in *non-interdependent* settings. In such settings, the outcome an actor A gets exclusively depends on A's own unilateral decisions, for example, how many resources to allocate to another person or a group. A's outcome, the consequences of the decision, does not depend on the choices of other agents, and hence, A's behavior is not subject to strategic considerations like concerns over reciprocity. Donating money to a noble cause⁹ is a prototypical example of prosocial behavior in non-interdependent settings.

One of the most prominent experimental paradigms for non-interdependent settings is the dictator game¹⁰ (DG). In the DG, a decision maker (i.e., the dictator) receives an endowment, and has to decide how to split this endowment between herself and another anonymous participant (i.e., the recipient). Because the recipient is powerless, the situation is non-interdependent, that is, the payoff the dictator receives is only dependent on the split of the endowment that she suggested. As expectations of reciprocity do not play a role in the DG, the dictator's behavior is a measure of pure prosociality^{11, 12}. What factors influence the decision to keep versus share resources in such non-interdependent settings? As a relevant psycho-physiological influence, hunger might bring about a focus on the immediate self-interest and might, therefore, reduce prosociality in non-interdependent settings. Some studies have started to address this question. Indeed, there is preliminary evidence in support of the idea that hunger increases selfishness¹³⁻¹⁵."

[...]

“As summarized above, hunger seems to reduce prosociality in non-interdependent settings, though the evidence may be considered tentative rather than conclusive. Does hunger also undermine prosociality in *interdependent* settings? In such settings, the outcome an actor A gets does *not* exclusively depend on A’s unilateral decision but depends on the choices of other agents. Hence, when they are mindful of their own outcomes, in interdependent situations, individuals cannot simply monitor only their own behavior, but have to consider key aspects of the social context, for example, their beliefs about the choices other people will make. A prominent experimental paradigm for interdependent settings is the Ultimatum Game (UG)¹⁸. The UG is an experimental game with two players. One player, the proposer, receives an endowment and she has to decide how to split this endowment between herself and the second player, the responder. The responder, in turn, has to decide whether she accepts the offer or rejects it. If the offer is accepted, the money is paid out accordingly. In case of rejection, both players receive nothing. (For an overview of a variety of non-interdependent and interdependent tasks and their payoff structures, see Fehr & Schmidt¹⁹; Kelley et al.²⁰).

In interdependent settings, the choice whether to act prosocial or not also entails strategic considerations and beliefs about the other person’s response²⁰⁻²². In several cases, humans can compensate for their own psycho-physiological impairments when the social context requires them to²³. This can be the case, for example, when the social situation elicits greater mental activity that is needed for more strategic forms of decision-making, such as ones involving beliefs about the likely behavior of other people. Hence, it is possible that in interdependent situations, people suppress any potentially selfish tendency triggered by hunger and, therefore, are less likely to decrease prosociality. To the best of our knowledge, only one study¹⁷ has investigated the effects of hunger in interdependent settings. In this study, Rantapuska and colleagues¹⁷ investigated the effects of experimentally manipulated hunger in two cooperation paradigms. Their study yielded results that were somewhat inconclusive, with increased prosociality in one of the tasks and a null finding in the other one, thus emphasizing the need for further research.”

In doing so, we made a clearer distinction between non-interdependent tasks (such as dictator games or SVO) and interdependent tasks (such as the ultimatum game), see p. 6-7:

“As non-interdependent measures we used the DG, social value orientation²⁴⁻²⁶, social mindfulness²⁷ and a volunteering task²⁸. Social value orientation (SVO) represents a person’s preference for (hypothetical) distributions of money between herself and another person in a set of non-constant-sum DGs^{29, 30}. More precisely, participants have to decide on a series of DG-like decisions, with different endowments and different distribution options. The SVO measure extends rational self-interest by simultaneously measuring the value people assign to other people’s outcomes^{25, 31}. SVO has been validated to be predictive of real-life prosocial behavior, such as donations to noble causes^{9, 32}, and volunteering^{28, 33}, as well as costly cooperation in economic games³⁴⁻³⁷. Social mindfulness²⁷ refers to the extent to which an individual’s decisions leave other people in decisional control, thereby respecting other people’s interest to choose freely for themselves. Behaving socially mindful can, therefore, be understood as a prosocial act, as it ensures options for the other person rather than removing options. It has been conceptualized as “low-cost cooperation”^{27, 38}. Volunteering^{28, 33} encompasses the investment of time resources to benefit others. As interdependent tasks, we used the UG, the Public Goods Game^{39, 40}, and the Stag Hunt Game⁴¹. In the Public Goods Game (PGG), multiple players can decide whether or not to contribute to a common pool that is afterwards multiplied by a constant and then split equally among all players. The prosocial choice is to contribute everything to the common pool as it increases the joint outcome. However, freeriding is possible, as the individual outcome can be maximized by keeping everything for oneself and still profiting from others’ contributions to the common pool⁴⁰. In the Stag Hunt Game (SHG), two players simultaneously decide between a cooperative, high pay-off, but socially risky option (‘hunting a stag’) and an uncooperative, low-pay-off, but safe option (‘hunting a hare’). Hunting the stag, the prosocial choice, results in a higher pay-off, but only if both players decide to hunt the stag. The hare, in contrast, can be hunted down independent of the decisions of the other player, but results in a lower payoff.”

Moreover, we discuss the validity of the SVO measure, see p. 6:

“The SVO measure extends rational self-interest by simultaneously measuring the value people assign to other people’s outcomes^{25, 31}. SVO has been validated to be predictive of real-life prosocial behavior, such as donations to noble causes^{9, 32}, and volunteering^{28, 33}, as well as costly cooperation in economic games³⁴⁻³⁷.”

Regarding your interpretation that the overall evidence points into the direction that hunger increases SVO, we are afraid that this might be a misunderstanding due to our presentation of the results: in Figure 1 values above zero represent *decreased* SVO when hungry. We now edited the caption of Figure 1 to avoid misunderstandings (see p.15):

“Figure 1. Effect sizes d of the effects of hunger on prosociality. Values above zero indicate effects in favor of a hypothesis that hunger decreases prosociality. Negative values indicate increased prosociality when hungry. Non-interdependent tasks: SVO, Social Mindfulness, DG, Volunteering. Interdependent tasks: PGG, SHG, UG. Effect sizes were calculated using formulas provided by Lipsey and Wilson⁵². Error bars indicate 95% confidence intervals.”

4) *It is unclear from the manuscript how the power analysis for Study 4 is conducted. The power analysis refers to a “conventionally medium size effect of $f = 0.25$ ” but what exact effect is this expected to reflect? The experimental is a factorial 2×2 design. Is it the interaction between the experimental factors, which is expected to equal .25, or is the main effect of hunger (i.e., averaging across the food vs. money conditions)? If it is the interaction, does the power analysis then account for the fact that an interaction is harder to detect than a main effect? (due to the higher standard errors that necessarily occurs because of the multicollinearity between interaction terms and the main terms).*

We appreciate your question regarding the power analysis. For two of the three dependent variables (volunteering task and SVO) we exclusively tested main effects. Also, for the dictator game, the potential moderation of effects of hunger by type of incentive was tested in a more exploratory manner, and again the main effect was in the focus of interest. When revising the manuscript, however, we became aware that we made a mistake in the calculation of the required sample size, which yielded a higher sample size than actually needed to detect an effect of $f = .25$, with corrected alpha: .017, and Power: .95. Hence, in the revised manuscript we now report a sensitivity analysis (similar to what we did in Study 3, where the participant pool was fixed). In this analysis, we calculated the required effect size given our sample of $N = 363$ (and alpha: .017, Power: .95). This effect size was even somewhat lower ($f = .21$), as compared to the a priori assumed effect of $f = .25$, see p. 26:

“A sensitivity analysis (using G*Power 3.1.9.2⁶²) revealed that, given our sample size of $N = 363$, alpha of .017 (Bonferroni-corrected), and power of .95, we could detect a small to medium size effect of $f = 0.21$ (main effect for hunger).”

In addition to these major concerns, I have the following minor comments:

5) *Study 4 reveals a significant effect of hunger on self-control. To me this seems to provide an obvious explanation for the lay theory that hunger increases self-interest. People (correctly) feel less able to control impulses when hungry and self-interest is certainly a strong impulse. What they might be forgetting is that, in the context of social interaction, there are a range of other countervailing impulses at play such as fairness or reputation. From an adaptationist perspective, such factors are just as ‘hard-wired’ (and, hence, impulse-based) as pure self-interest. If the authors agree, the finding related to self-control might be worthwhile to highlight in the discussion.*

This is a great idea! We added a section in the discussion on countervailing impulses as you suggested. Here, we argue that this might even be a foundation of the lay theories that hunger increases selfishness, see p. 19:

“The interpretation that hunger increases egoistic impulses that are suppressed by social requirements is in line with countervailing direct and indirect effects found in Aarøe and Peterson’s¹⁶ study. It is also in line with our finding in Study 4 that hunger significantly reduces perceived self-control, which does not translate into more selfish behavior. People might perceive themselves to be less able to control impulses when hungry but, in the context of social interaction, there is a range of other countervailing impulses at play, such as fairness or risk of rejection. The experience of lowered self-control when hungry might even be a foundation of the lay theories that hunger decreases prosociality, as shown by study 5. Speculatively, people might be able to accurately judge the impact that hunger has on their self-control, but unable to correctly anticipate the influence of social factors that maintain prosociality.”

6) *To help the reader assessment the strength of the underlying evidence, I think it would be helpful to have the N s of each individual study when the results of the studies are described and also provide more elaboration on the nature of experimental manipulations. Everything is in the methods section*

but, presently, it is difficult to assess the results without going back and forth between the results and the methods section.

We agree that more information about sample sizes, designs and measures before presenting the results would be helpful (Reviewer 2 had a similar point, see Reviewer 2 point 3). In the revised version of the manuscript, we now, (a) present an overview of the measures in the theory section (see p. 6-7), (b) extend the description of the study designs and manipulations in the overview of the studies (see p. 8, 11, and 16), and (c) include a table that provides an overview of the four studies, Ns, designs, measures (see table 1, p. 7).

7) *The authors write: “The data that support the findings of this study are available from the corresponding author upon reasonable request.” Given the authors’ laudable commitment to open science (as reflected in the use of pre-registration), I was curious why they wouldn’t also just share the data and analyses files in a public repository?*

Yes, we thought carefully about this issue before submitting, and decided that we will do this once the manuscript is accepted for publication. We have already changed the data availability statement accordingly (see p. 29).

Reviewer #2

This paper investigates whether hunger affects prosocial behavior. Several experimental and non-experimental methods are used to create variation in hunger. Several instruments are used to assess prosociality. Overall, no effect is found. This implies that the relevance of the study has to be assessed based on the relevance of the research question. I consider it as relevant because there is empirical evidence for the existence of a hunger effect. In addition, the authors show that lay people believe that hunger reduces prosociality.

The hunger induction is well done. In Study 1 and 2 participants are recruited in a hungry state and half of them get food; in Study 3 a correlational approach based on subjective hunger is used; and in Study 4 students are recruited before and after lunch. It is also positive that Study 3 and 4 are rather high powered and preregistered.

Thank you for your positive evaluation of our work.

General Comments:

1. *The measurement of prosociality is based on several games. However, only social value orientation and the dictator game provide undistorted measures of prosociality. In the other games, the behavior of the player does not only depend on the prosociality but also on how the participants assess the behavior of the other players. This is obvious in the ultimatum game, the trust game and the stag hunt game where also selfish players behave prosocially if they believe in the prosociality of the other player(s). Vice versa, in the public goods game, also prosocial (conditionally cooperative) players behave as selfish players if they believe that the other players are selfish. This makes the analysis difficult. It is not so clear that the different measures measure the same concepts. So it is not clear how the evidence should be aggregated. The authors use a Bonferroni correction in order to take multiple comparisons into account. This is safe not to overestimate significance, but it is not the best argument for a null result. Actually, the integrated measure that considers only the SVO, which is truly prosocial behavior, shows a significant effect in the expected direction. I wonder whether a principal component analysis or a method using latent concepts could replicate the “established” result.*

Thank you for raising this very important point! Reviewer 1 had a similar point regarding the validity of the different measures used in the different studies (see Reviewer 1, point 3). We absolutely agree that particularly the dictator game and the social value orientation scale (which, in our studies, was a slider measure with a series of non-constant-sum dictator games) provide undistorted measures of prosociality. However, note that our measure of social mindfulness is an indicator of “pure” prosociality (and has been conceptualized as such; e.g., Van Doesum et al., 2013, JPSP). Also, the volunteering task used in Study 4 can be considered a measure of prosociality. In contrast, we agree that the ultimatum game (Study 2), the public goods game (Study 1 & 2), and the stag hunt game (Study 1) do not exclusively measure prosociality but also have strategic components and entail social risk taking (see e.g., Balliet, Wu, & De Dreu, 2014; Rand, 2016, Psychological Science). To enhance clarity and provide a more comprehensive rationale, we made four major revisions.

First, in the revised version of the manuscript, we rewrote the whole theory section to clarify the nature of the measures used in the studies, see p. 3-7.

In doing so, we made a clearer distinction between non-interdependent tasks (such as dictator games or SVO) and interdependent tasks (such as the ultimatum game), see p. 6-7.

Second, in the revised manuscript, we included a table with an overview of the four studies, and the various measures included in each study (see Table 1, p. 7). The goal is to make the reader aware of the variety and breadth of the measures of prosociality that we included in our research.

Third, we really liked your idea of a better aggregation of the data and the revision of the theory section guided us how to do it. Unfortunately, the use of a latent variable approach is not suitable for our studies, because each task presents one item and different tasks were used in different studies. Only Study 2 contained clearly non-interdependent tasks (SVO) and interdependent tasks (Ultimatum Game, Public Goods Game). The limited number of manifest variables to measure each latent construct (i.e., only one item for the non-interdependent context) and the relatively small sample size in Study 2 does not allow for a CFA. Instead, we followed your advice and carried out a principal component analysis for Studies 2 and 4, because these studies measured SVO together with other measures of prosocial behavior (Study 3 only contained a measure of SVO, but no other measures, and Study 1 did not include SVO or a dictator game).

For Study 2 we found, as expected, that SVO prosocial behavior shares variance with social mindfulness and the one-shot public good game behavior. Furthermore, Ultimatum game proposals shared variance with public good game behavior, but not the measures of unconditional prosociality (see results of the PCA in Table 1).

Table 1 Results of PCA for Study 2.

	Component		Uniqueness
	1	2	
UG offer		0.887	0.199
Social mindfulness	0.724		0.413
SVO_angle	0.822		0.273
PGG contribution	0.504	0.629	0.351

Note. 'varimax' rotation was used

In Study 4, all measures of prosocial behavior were entered into the PCA. All three measures, SVO, dictator game giving and volunteering load on one component. Importantly, albeit correlated, volunteering seems unique (see results of the PCA in Table 2).

Table 2 Results of PCA for Study 4.

	Component		Uniqueness
	1		

Component Loadings		
	Component	
	1	Uniqueness
SVO_angle	0.701	0.509
Dictator Game	0.780	0.391
Time volunteering	0.395	0.844

Note. 'varimax' rotation was used

Fourth, in the revised manuscript, we recalculated the integrated effect sizes. We now report an integrated effect size for all DVs, but also separate effect sizes for the non-interdependent tasks (SVO, dictator game, volunteering, social mindfulness) and the interdependent tasks (stag hunt game, public goods game, ultimatum game), see p. 15. Due to the re-aggregation, we now found a small, though significant integrated effect ($d = 0.108$) for the non-interdependent task, but no significant integrated effect for the interdependent tasks ($d = -0.073$) and discuss this finding on p. 17:

“The integrated effect size including all DVs from our four studies was close to zero ($d = 0.076$) and the 95% CI included zero [-0.024; 0.169]. The same was true for the integrated effect size for interdependent tasks ($d = -0.073$, CI [-0.296; 0.143]). There was a very small integrated effect size of $d = .108$, 95% CI [0.001; 0.216] for non-interdependent tasks. Hence, the overall picture does not point to a general or pronounced detrimental effect of hunger on prosociality. Only for the integrated non-interdependent tasks there was, however weak, evidence for the prediction that hunger decreases prosociality.”

Regarding the Bonferroni correction, we deem it adequate given the relatively high number of outcomes/dependent variables used in our studies. The decision whether or not to control for inflation of type I errors should – in our eyes – be independent of whether null hypotheses are tested or alternative hypotheses. Moreover, please note that we did not explicitly test a null hypothesis, but our tentative prediction building on previous studies and theorizing was that hunger might have an effect on prosociality. Finally, the effect sizes that are of central interest in case of our paper (as we calculated integrated effect sizes combining evidence from different studies) are not influenced. For these three reasons, we decided to retain the Bonferroni correction. Of course, if you have another suggestion, we are happy to consider it.

2. In the ultimatum games deception was used. It is known that deception can reduce participants' credibility in the experimenter. However, I am aware of the fact that this is not considered as a problem in psychology. In the dictator game the participants could donate to an anonymous future participant. It is not clear how this is done. It is important to know whether the recipients were also in the role of the dictator because in this case no transfer by all dictators is also fair.

We agree that deception in experimental studies has its drawbacks and that it can reduce the experimenter's credibility. However, for practical reasons (i.e., immediate pay-offs after the experimental session), we regarded the small form of deception used here as reasonable. At least two reasons are important. First, it is unlikely that it harms the participants in any important manner, especially given decades of research on economic games and prosociality in which often stronger forms of deception were used. Second, and more importantly, the participants had no prior experience with games from behavioral economics (this was an exclusion criterion). So, if anything, the reduced credibility might become a problem for future studies, but did not affect the present findings.

Finally, we should note that our studies also include actual money. For example, the donations made in the dictator game were paid out to subsequent participants, after their participation. Prior to participation, participants were not informed that they would receive additional money (donated by previous participants) at the end of the experiment. Moreover, participants were not informed that the future participants (they were donating money to) would also be in the role of a dictator. We added a sentence on this issue in the description of the dictator game, see p.27:

“The donations made in the DG were paid out to subsequent participants, after their participation. Prior to participation, participants were not informed that they would receive additional money at the end of the experiment. Moreover, participants were not informed that the future participants would also be in the role of a dictator.”

3. *It is fine to separate methods from the main text. However the main text should contain the all relevant information. For example, it is admittedly natural to induce hunger by giving food to the control group. Explicitly mention this in the text would help the reader to immediately understand that the treatment group (hunger) is the untreated group.*

We agree that more information about sample sizes, designs and measures before presenting the results is very helpful for the reader (Reviewer 1 had a similar point, see Reviewer 1, point 6). In the revised version of the manuscript, we (a) present an overview of the measures in the theory section (see p. 6-7), (b) extend the description of the study designs and manipulations in the overview of the studies (see p. 8, 11, and 16), and (c) include a table that provides an overview of the four studies, Ns, designs, measures (see table 1, p. 7).

Assessment

The paper addresses a relevant research question with a comprehensive study. It produces a null result but this null result will not be undisputed. I recommend inviting a revision, which improves clarity in the main text, provides a better theoretical foundation of the different measures, and analyses the data more deeply.

Thank you very much for the positive evaluation. We also believe that the null findings are informative and will contribute to a lively debate.

Specific comments

4. *The paper should be clearer in the discussion of the measures of prosociality (see above). For example, consider economic papers that game theoretically analyze these games assuming non-selfish preferences as Ernst Fehr and Klaus M. Schmidt. 1999. "A Theory of Fairness, Competition, and Cooperation." Quarterly Journal of Economics, 114 3, 817-68.*

Provide also reference to the original papers of the different game (UG, trust game...).

Thank you for this suggestion. When rewriting the theory section we devoted particular attention to the different measures of prosociality, both its conceptual meaning (e.g., the distinction between unilateral forms of prosociality in non-interdependent settings, and strategic forms of prosociality in interdependent settings), as well as their validity in various contexts, thereby recognizing key articles that have in many ways shaped the study of prosociality (see p. 3-7):

“It has been argued that hunger, as a signal of limited resource availability, reduces prosociality, that is, the willingness to invest one’s own resources (e.g., money, time, effort) to help others^{7,8}. Such “pure prosociality” is typically measured in *non-interdependent* settings. In such settings, the outcome an actor A gets exclusively depends on A’s own unilateral decisions, for example, how many resources to allocate to another person or a group. A’s outcome, the consequences of the decision, does not depend on the choices of other agents, and hence, A’s behavior is not subject to strategic considerations like concerns over reciprocity. Donating money to a noble cause⁹ is a prototypical example of prosocial behavior in non-interdependent settings.

One of the most prominent experimental paradigms for non-interdependent settings is the dictator game¹⁰ (DG). In the DG, a decision maker (i.e., the dictator) receives an endowment, and has to decide how to split this endowment between herself and another anonymous participant (i.e., the recipient). Because the recipient is powerless, the situation is non-interdependent, that is, the payoff the dictator receives is only dependent on the split of the endowment that she suggested. As expectations of reciprocity do not play a role in the DG, the dictator’s behavior is a measure of pure prosociality^{11,12}. What factors influence the decision to keep versus share resources in such non-interdependent settings? As a relevant psycho-physiological influence, hunger might bring about a focus on the immediate self-interest and might, therefore, reduce prosociality in non-interdependent settings. Some studies have started to address this question. Indeed, there is preliminary evidence in support of the idea that hunger increases selfishness¹³⁻¹⁵.”

[...]

“As summarized above, hunger seems to reduce prosociality in non-interdependent settings, though the evidence may be considered tentative rather than conclusive. Does hunger also undermine prosociality in *interdependent* settings? In such settings, the outcome an actor A gets does *not* exclusively depend on A’s unilateral decision but depends on the choices of other agents. Hence, when they are mindful of their own outcomes, in interdependent situations, individuals cannot simply monitor only their own behavior, but have to consider key aspects of the social context, for example, their beliefs about the choices other people will make. A prominent experimental paradigm for interdependent settings is the Ultimatum Game (UG)¹⁸. The UG is an experimental game with two players. One player, the proposer, receives an endowment and she has to decide how to split this endowment between herself and the second player, the responder. The responder, in turn, has to decide whether she accepts the offer or rejects it. If the offer is accepted, the money is paid out accordingly. In case of rejection, both players receive nothing. (For an overview of a variety of non-interdependent and interdependent tasks and their payoff structures, see Fehr & Schmidt¹⁹; Kelley et al.²⁰).

In interdependent settings, the choice whether to act prosocial or not also entails strategic considerations and beliefs about the other person’s response²⁰⁻²². In several cases, humans can compensate for their own psycho-physiological impairments when the social context requires them to²³. This can be the case, for example, when the social situation elicits greater mental activity that is needed for more strategic forms of decision-making, such as ones involving beliefs about the likely behavior of other people. Hence, it is possible that in interdependent situations, people suppress any potentially selfish tendency triggered by hunger and, therefore, are less likely to decrease prosociality.” [...]

“As non-interdependent measures we used the DG, social value orientation²⁴⁻²⁶, social mindfulness²⁷ and a volunteering task²⁸. Social value orientation (SVO) represents a person’s preference for (hypothetical) distributions of money between herself and another person in a set of non-constant-sum DGs^{29, 30}. More precisely, participants have to decide on a series of DG-like decisions, with different endowments and different distribution options. The SVO measure extends rational self-interest by simultaneously measuring the value people assign to other people’s outcomes^{25, 31}. SVO has been validated to be predictive of real-life prosocial behavior, such as donations to noble causes^{9, 32}, and volunteering^{28, 33}, as well as costly cooperation in economic games³⁴⁻³⁷.

Social mindfulness²⁷ refers to the extent to which an individual’s decisions leave other people in decisional control, thereby respecting other people’s interest to choose freely for themselves. Behaving socially mindful can, therefore, be understood as a prosocial act, as it ensures options for the other person rather than removing options. It has been conceptualized as “low-cost cooperation”^{27, 38}. Volunteering^{28, 33} encompasses the investment of time resources to benefit others.

As interdependent tasks, we used the UG, the Public Goods Game^{39,40}, and the Stag Hunt Game⁴¹. In the Public Goods Game (PGG), multiple players can decide whether or not to contribute to a common pool that is afterwards multiplied by a constant and then split equally among all players. The prosocial choice is to contribute everything to the common pool as it increases the joint outcome. However, freeriding is possible, as the individual outcome can be maximized by keeping everything for oneself and still profiting from others’ contributions to the common pool⁴⁰. In the Stag Hunt Game (SHG), two players simultaneously decide between a cooperative, high pay-off, but socially risky option (‘hunting a stag’) and an uncooperative, low-pay-off, but safe option (‘hunting a hare’). Hunting the stag, the prosocial choice, results in a higher pay-off, but only if both players decide to hunt the stag. The hare, in contrast, can be hunted down independent of the decisions of the other player, but results in a lower payoff.”

Thank you for bringing our attention to this classic paper that is indeed very helpful! We included this reference when providing an overview of the different tasks we used in our studies (p. 5) and we also included the original references for the different games (e.g., p. 3, 5, and 7).

5. *The analysis should take the nature of the different games into account (see above).*

As described above, in our analyses we now stronger account for the different nature of the games (see p. 15 and Figure 1).

6. *Page 3, „Keeping versus using resources is particularly relevant for cooperation, as cooperating with others requires investing one’s own resources (e.g., money, time, effort) to achieve collective goals.” Keeping versus using resources is relevant in all dimensions of prosocial behavior. Actually, most definitions of prosocial behavior assume that prosocial requires giving up some resources in order help others at a cost.*

We agree. This sentence was deleted during the revision.

7. Page 5. *Trust is considered as non-interdependent measure of prosociality. I disagree with this view. Trust is highly interdependent. It relies on the others' trustworthiness. So it depends on the expected trustworthiness of the trustee.*

We completely agree. As no clear prediction could be made for trust, and as the self-report measure via Likert scale is fundamentally different from the measures of prosociality on the basis of resource allocation decisions, we decided to no longer present these findings in the manuscript. Particularly, we no longer include these effect sizes in the integrated effect sizes. However, in the "Exploratory measures" section, we announce that trust was measured (see p. 25).

8. Page 6. *In the results the order of the treatments varies across the different measure. This makes it more difficult to read.*

Thank you for this suggestion. We now present them in an identical order.

9. Page 10, *presentation of the dictator game. Report the all four transfers of the 2x2 design and check separately whether hunger affects decisions in the money and food treatment. Alternatively use a regression.*

On p. 13, we now report the transfers for all four conditions. We did not report simple effects of hunger in the food and the money condition, because the hunger x type of incentive interaction was not significant ($p = .850$). In line with this, both simple effects are not significant (monetary incentive condition: $p = .724$; food incentive condition: $p = .494$). If you think it would be important to report the simple effects, despite the non-significant interaction, we will readily do so.

10. *In the lay study, the correctness of the guess was not incentivized. It could cause the participants to respond more in line with an experimenter demand effect.*

Indeed, correctness was incentivized, and we presented this information in the method section (see p. 28):

"They received an additional bonus of \$0.50 for accurate estimates (i.e., within a 20% range around the actual result for the hungry group)"

In addition, we now included this information in the overview of Study 5, see p. 16:

"The correctness of guesses was incentivized."

Typos

page 5. *"non-interdepended"*

Corrected

Check again the references. The second author in reference 19 is Kurzban and not Kurzbahn.

Corrected

Reviewer #3:

1. *The hypothesis is well worth testing, but I do not believe these studies are able to really address the issue. The primary problem is hunger in the context of these studies is just a minor inconvenience. Having not eaten since 10pm the previous evening is unlikely to be a major issue for most subjects, especially given they know that they can feed in a short period of time. This is particularly true for those just about to enter the cafeteria. This form of hunger is nothing like the hunger someone experiences if they are suffering deprivation due to a true food shortage brought on by a drought or other catastrophic events. I believe the null result is because the treatment is just too weak to have any significant effect on behavior. I do not think we have learned anything.*

Thank you very much for raising this important concern! Discussions amongst us and re-thinking the phrasing of our manuscript in light of your criticism sharpened which questions our paper can – and which it cannot – answer. Overall, we agree with you that it might be difficult to generalize from our studies to situations in which hunger poses an existential threat. Hence, we completely understand your concern, and therefore included a discussion about the generalizability of the findings. Indeed, our findings do not generalize to more extreme or chronic forms of hunger, or even contexts in which poverty is common (see below).

At the same time, we regard our findings to be of great interest for the following reasons. First, our primary goal was to replicate, and extend, previous research examining the effects of *acute* hunger– and this research reported significant effects. Importantly, the present manipulations of hunger are at least equally strong to those used in the previous studies that have been published (e.g., Briers, Pandelaere, Dewitte, & Warlop, 2006; Harel, & Kogut, 2015; Rantapuska, Freese, Jääskeläinen, & Hytönen, 2017; Xu, Bègue, Sauve, & Bushman, 2014). In terms of reproducibility and generalizability, we therefore think that our null findings are highly informative (in particular, in times of the replicability crisis in psychology).

Second, our manipulations of hunger capture acute hunger rather than chronic hunger. As such, we believe that our findings are relevant to situations in which people have acute hunger, such as before lunch or dinner, or perhaps when travelling or other circumstances somehow prevent one from access to food. Most people in industrialized societies have experience with such forms of acute hunger. Therefore, while limited to certain societies, the present findings pertain to realistic situations with which most of our participants have some experience.

Third, Study 5 uncovered that lay theories do state that the forms of acute hunger that we manipulated or examined undermine prosociality (and note that we used incentivized procedures of assessing these beliefs). This is another reason why we believe that the present findings obtained for acute hunger (rather than chronic hunger) *are* newsworthy – at least for many or most people.

It is also important to note some methodological, practical, and ethical limitations that make it hard, or virtually impossible, to examine the clean effects of severe hunger. First, examining hunger in the field would entail a high likelihood of strong confounds, for example socioeconomic status, cultural differences, or acute threats due to natural disasters, all of which have been found to be related to prosociality in their own right (Bell et al., 2009; Piff et al. 2010; Whitt & Wilson, 2007). These confounds might overshadow potential effects of hunger, or on the contrary, produce spurious correlations. Second, in experimental research there are practical, ethical, and methodological issues when one seeks to manipulate extreme or chronic hunger in a world where this is not common. For example, it is likely that such research raises serious ethical issues (our universities would most likely disapprove of inducing extreme or chronic hunger) as well as questions about validity as it may activate many other psychological processes. For example, extreme hunger may be associated with feelings of injustice, anger (directed at the experimenters), and perhaps other strong emotions or negative beliefs about the experiment. Finally, one is very likely to obtain a non-random sample if extreme hunger is announced as a key feature of the study.

In the revised manuscript, we addressed your concern in three ways: first, throughout the whole manuscript, particularly the theory section (that we have fully rewritten), we now refrain from using phrases that imply associations with chronic or severe hunger (e.g., hardship, catastrophes, scarcity). In the theory section, we also define acute hunger, which is the focus of our research (see p.3);

“Acute hunger due to temporary deprivation of food is characterized by craving for food, feeling hungry, and, on a physiological level, decreased blood glucose levels.”

Second, in the discussion, we now included a longer paragraph (see p. 20-21) clarifying the scope of the paper (acute hunger). And third, we briefly discuss to what extent the present findings might generalize to more extreme forms of hunger:

“Acute versus chronic hunger

To avoid misunderstandings, note that we examined hunger in terms of an acute condition within the upper boundaries of the natural daily fluctuation typical in Western industrialized societies (Studies 1 to 4 were conducted with undergraduate student samples in Germany). The effects of hunger in terms of a chronic, potential life-threatening state, as a consequence of food scarcity due to poverty, natural disasters, or war is beyond the scope of the present paper. Rather, our primary aim was to replicate and extend previous research and theorizing on the effects of acute hunger and/or blood glucose on prosociality¹³⁻¹⁷. All of these studies used manipulations similar to ours or weaker, for example, the application of a carbonated versus non-carbonated soft drink without prior food deprivation^{14, 16}. Most of these studies reported an effect of hunger or blood glucose on prosociality. Hence, in terms of reproducibility and generalizability, as well as in terms of advancement of theory development, we therefore deem our new evidence in support of the null hypothesis (7 of 12 tests) to be highly informative.

As noted earlier, we are fully aware of the qualitative difference between acute hunger as a non-critical state and chronic hunger as an existential threat. Nevertheless, we believe that our manipulations assess non-trivial forms of hunger, as many people in Western industrialized worlds might experience them. For example, our variations in hunger were validated by strong and consistent effects on two key aspects of hunger: blood glucose levels (measured in Studies 1 and 2), and self-reported hunger (measured in Studies 1, 2, and 4). As natural fluctuations in blood glucose are regulated within a narrow range, even short food deprivation is likely to cause blood glucose levels to be at the lower boundary of the regulatory corridor^{58, 59}. Notably, we did not detect any significant link between levels of blood glucose and degree of prosociality in Studies 1 and 2.

Although it is beyond the scope of this paper, we believe that – given the strength of our manipulations and the small range of natural blood glucose levels – our findings provide a first indication that also severe forms of hunger might not *per se* undermine prosociality. From an evolutionary perspective, it could be argued (albeit speculatively) that shortage of food should even foster prosociality and cooperation. For example, Gurven⁶⁰ emphasized that in small scale societies meeting daily food needs is a highly interdependent task. Interestingly, in such societies scarce food resources even increase prosociality (peaceful sharing) rather than selfishness. This tendency might be hard-wired as a consequence of mankind's long history of cooperative hunting and gathering. Evidence from evolutionary game theory⁶¹ supports the idea that in times of scarcity (i.e., rare pay-offs), evolution favors strategies that minimize the risk of not receiving any resources at all rather than strategies that maximize resources, which also relates to the idea that, when facing limited resources, people might strive to cooperate.”

Minor issues:

2. For the non-social risk task, just counting the number of risking choices can mask problems with this type of risk attitude measure. It is not uncommon in tasks such as this for subjects to make inconsistent choices. The authors do not address this issue. Do their subjects make inconsistent choices?

Thank you for bringing this issue to our attention. The main purpose of the non-social risk task was to safeguard the specificity of a potential effect of hunger on prosociality. Hence, the non-social risk task was used to rule out an alternative explanation that hunger affects risk preferences in general (i.e., also in non-social context). However, as we found no effects of hunger on prosociality, the non-social risk task became a bit pointless (and we omitted it from the follow-up studies 3-4). In the revised manuscript, we therefore decided not to report the non-social risk task and its results in detail, but to only announce that we included it (see p. 23). We deem this to be most adequate as the main purpose for the inclusion of the task was void.

We also agree that inconsistencies in risk choices are not uncommon (and interesting). When scoring the non-social risk task we followed the procedure of von Dawans et al. (2012), from whom we adapted the task and who also simply summed the scores. We also had no explicit hypotheses regarding how and why hunger should affect (in)consistency in risk preferences. Nonetheless, in an exploratory manner, we now analyzed inconsistent choices in the non-social risk tasks. In total, 7 of the 62 (11%) participants made at least one inconsistent decision in the non-social lotteries (i.e., they chose the risky option in one lottery but the non-risky option in a subsequent lottery, although in the latter lottery the expected value for the risky option was higher as compared to the former lottery, with identical non-risky options in both lotteries). The distribution of these participants did not significantly differ between experimental conditions (hungry: 4 out of 31, not hungry: 3 out of 31).

3. *The ultimatum game is not a good measure of prosocial attitude. Keeping more for yourself is a risky decision; the probability of a rejection increases with the amount kept. So a more generous offer may reflect risk aversion rather than prosociality.*

Thank you very much for this important pointer. In the revised manuscript, we clarified the operationalizations and validity of our measures. We fully agree that the decisions made in the ultimatum game do not (exclusively) reflect prosociality, but are strongly influenced by strategic considerations and risk preferences, see p. 5:

“As summarized above, hunger seems to reduce prosociality in non-interdependent settings, though the evidence may be considered tentative rather than conclusive. Does hunger also undermine prosociality in *interdependent* settings? In such settings, the outcome an actor A gets does *not* exclusively depend on A’s unilateral decision but depends on the choices of other agents. Hence, when they are mindful of their own outcomes, in interdependent situations, individuals cannot simply monitor only their own behavior, but have to consider key aspects of the social context, for example, their beliefs about the choices other people will make. A prominent experimental paradigm for interdependent settings is the Ultimatum Game (UG)¹⁸. The UG is an experimental game with two players. One player, the proposer, receives an endowment and she has to decide how to split this endowment between herself and the second player, the responder. The responder, in turn, has to decide whether she accepts the offer or rejects it. If the offer is accepted, the money is paid out accordingly. In case of rejection, both players receive nothing. (For an overview of a variety of non-interdependent and interdependent tasks and their payoff structures, see Fehr & Schmidt¹⁹; Kelley et al.²⁰).

In interdependent settings, the choice whether to act prosocial or not also entails strategic considerations and beliefs about the other person’s response²⁰⁻²². In several cases, humans can compensate for their own psycho-physiological impairments when the social context requires them to²³. This can be the case, for example, when the social situation elicits greater mental activity that is needed for more strategic forms of decision-making, such as ones involving beliefs about the likely behavior of other people. Hence, it is possible that in interdependent situations, people suppress any potentially selfish tendency triggered by hunger and, therefore, are less likely to decrease prosociality.”

Also, in the discussion section, we return to this issue, see p. 18-19:

“Moreover, people might underestimate the social context that dictates cooperation in some cases, even if self-interest was triggered. This might be particularly true for situations that represent interdependence, as in such situations hunger could not directly translate into selfishness. Our analyses revealed a – very small – effect of hunger on prosociality in non-interdependent contexts; that is, when the outcome an individual receives depends exclusively on her decisions. These are prototypical situations in which giving in to selfish impulses is not believed to result in repercussions – such as rejection decisions in the UG or increased selfishness by others in interdependence tasks. Strikingly, even in such situations, which in our case were also stripped of elements of social resonance or non-anonymity, we found only a very small negative effect of hunger. We suggest that most conflicts between self-interest and prosociality in everyday life entail some degree of interdependence – a feature captured by the interdependent tasks used in our studies. For example, in an UG, participants cannot simply reduce their contributions to protect their resources without increasing the danger of eliciting a negative response by others that leads to receiving no rewards at all, when the offer is rejected. The null findings in these situations might indicate that, even if hunger increases the urge to acquire more resources, people are still able to consider their interdependence with others and the strategic constraints of the situation, which is in line with evidence that hunger even improves strategic decision making⁵⁵. In non-interdependent situations, people might at least be aware of social norms and might be guided by them in their behavior⁵⁶, as the non-significant findings and the only small integrated effect size for these tasks indicate. More generally, our findings are in line with the argument that people are able to compensate for their psycho-physiological impairments when they perceive it as required by the social context^{23, 57}.”

References

- Aarøe, L., & Petersen, M.B. (2013). Hunger Games: Fluctuations in Blood Glucose Levels Influence Support for Social Welfare, *Psychological Science*, 24, 2550-2556.
- Balliet, D., Wu, J., & De Dreu, C. K. (2014). Ingroup favoritism in cooperation: a meta-analysis. *Psychological bulletin*, 140(6), 1556.

- Bell, A. V., Richerson, P. J., & McElreath, R. (2009). Culture rather than genes provides greater scope for the evolution of large-scale human prosociality. *Proceedings of the National Academy of Sciences*, *106*(42), 17671-17674.
- Briers, B., Pandelaere, M., Dewitte, S., & Warlop, L. (2006). Hungry for money: The desire for caloric resources increases the desire for financial resources and vice versa. *Psychological Science*, *17*, 939-943.
- Harel, I., & Kogut, T. (2015). Visceral needs and donation decisions: Do people identify with suffering or with relief?. *Journal of Experimental Social Psychology*, *56*, 24-29.
- Piff, P. K., Kraus, M. W., Côté, S., Cheng, B. H., & Keltner, D. (2010). Having less, giving more: the influence of social class on prosocial behavior. *Journal of personality and social psychology*, *99*(5), 771.
- Rand, D. G. (2016). Cooperation, Fast and Slow: Meta-Analytic Evidence for a Theory of Social Heuristics and Self-Interested Deliberation. *Psychological Science*, *27*(9), 1192-1206.
- Rantapuska, E., Freese, R., Jääskeläinen, I. P., & Hytönen, K. (2017). Does Short-Term Hunger Increase Trust and Trustworthiness in a High Trust Society? *Frontiers in psychology*, *8*, 1944.
- Van Doesum, N. J., Van Lange, D. A. W., & Van Lange, P. A. M. (2013). Social mindfulness: Skill and will to navigate the social world. *Journal of Personality and Social Psychology*, *105*, 86-103.
- Von Dawans, B., Fischbacher, U., Kirschbaum, C., Fehr, E., & Heinrichs, M. (2012). The social dimension of stress reactivity: acute stress increases prosocial behavior in humans. *Psychological Science*, *23*, 651-660.
- Whitt, S., & Wilson, R. K. (2007). Public goods in the field: Katrina evacuees in Houston. *Southern Economic Journal*, *377-387*.
- Xu, H., Bègue, L., Sauve, L., & Bushman, B. J. (2014). Sweetened blood sweetens behavior. Ego depletion, glucose, guilt, and prosocial behavior. *Appetite*, *81*, 8-11.

Reviewer #2 (Remarks to the Author):

This revision addresses all issues raised in the report on the previous version. Most of the answers satisfy me and I consider this version as a clear improvement over the previous version.

Nevertheless there are some issues that should be addressed in a revision.

1. It is good to distinguish between the non-interdependent and the interdependent settings. Also the explanations of this difference are very helpful. However, the expression "prosociality in the interdependent settings" is strictly speaking not correct since it might be driven by other motives. This should be made clear when the term is used for the first time, for example in a footnote.
2. The explanations on line 99ff, which explain why the decisions in the interdependent settings are more difficult to interpret, leave the crucial argument out. Because the decision can influence the decision of the second mover, the motive for the decision of the first mover is ambiguous. In the case of the ultimatum game, a fair offer can be motivated by a fairness motive or by the fear that a lower offer is rejected. The explanations on line 110ff are confusing in this regard. It is not clear whether a selfish person has to suppress something when making a fair offer. The participant has just a tradeoff between a rather fair offer with a lower offer that comes with a higher rejection risk.
3. You write that you stick to the use of Bonferroni correction. I agree that this is the proper way to test the hypothesis. Nevertheless, this weakens the evidence for the null result. Since the procedure is transparent, the reader can form his or her own opinion. Nevertheless, I would for example suggest removing the word "any" in the sentence "we did not find any significant effects of hunger on prosociality in the individual DVs".
4. The trust game was excluded from the main analysis. In my view, it should be treated as the ultimatum game and included in the interdependent settings.
5. Concerning deception, I appreciate that there were recipients in the dictator games. This was my main concern because otherwise participants could feel deceived because the transfer was not made. However, I would like to note that the argument that deception would only affect future experiments is quite dangerous. Actually, it implies that it would be much more efficient to block this experiment than all future ones.

Reviewer #3 (Remarks to the Author):

For full transparency, the title should include the word "acute."

Regarding the counting the number of risking choices issue, the fact that someone else took this flawed approach and got it published, is not justification for continuing the flawed approach.

I am still not sold that the paper is really telling us anything. I still believe the null result is because the treatment is just too weak to have any significant effect on behavior.

REVIEWERS' COMMENTS:

Reviewer #1 (Remarks to the Author):

I thank the authors for dealing with my comments in such a detailed and constructive manner. I have no further concerns and believe that this manuscript makes an important contribution.

- Michael Bang Petersen

Thank you again for your very helpful feedback, and for the time and effort that you have devoted to our paper.

Reviewer #2 (Remarks to the Author):

This revision addresses all issues raised in the report on the previous version. Most of the answers satisfy me and I consider this version as a clear improvement over the previous version. Nevertheless there are some issues that should be addressed in a revision.

Thank you for this positive evaluation of our revised manuscript.

1. It is good to distinguish between the non-interdependent and the interdependent settings. Also the explanations of this difference are very helpful. However, the expression "prosociality in the interdependent settings" is strictly speaking not correct since it might be driven by other motives. This should be made clear when the term is used for the first time, for example in a footnote.

And

2. The explanations on line 99ff, which explain why the decisions in the interdependent settings are more difficult to interpret, leave the crucial argument out. Because the decision can influence the decision of the second mover, the motive for the decision of the first mover is ambiguous. In the case of the ultimatum game, a fair offer can be motivated by a fairness motive or by the fear that a lower offer is rejected. The explanations on line 110ff are confusing in this regard. It is not clear whether a selfish person has to suppress something when making a fair offer. The participant has just a tradeoff between a rather fair offer with a lower offer that comes with a higher rejection risk.

Thank you for this pointer. We agree and clarified these issues by reformulating large parts of the section introducing the difference between interdependent and non-interdependent settings of prosociality on p. 5/6:

*"As summarized above, hunger seems to reduce prosociality in non-interdependent settings, though the evidence should be considered tentative rather than conclusive. Does hunger also undermine prosociality in *interdependent* settings? In such settings, the outcome an actor A gets does *not* exclusively depend on A's unilateral decision but depends on the choices of other agents. Hence, when they are mindful of their own outcomes, in interdependent situations, individuals cannot simply monitor only their own behavior, but have to consider key aspects of the social context, for example, their beliefs about the choices other people will make. Hence, unlike non-interdependent settings, interdependent settings do not measure pure prosociality as the decisions of an actor A can be influenced by other motives, such as strategic concerns. A prominent experimental paradigm for interdependent settings is the Ultimatum Game (UG)¹⁸. The UG is an experimental game with two players. One player, the proposer, receives an endowment and she has to decide how to split this endowment between herself and the second player, the responder. The responder, in turn, has to decide whether she accepts the offer or rejects it. If the offer is accepted, the money is paid out accordingly. In case of rejection, both players receive*

nothing. As opposed to the DG, where the recipient of the endowment is powerless, in the UG, the proposer has to anticipate the reaction of the responder. Hence, a fair offer of the proposer in the UG can be motivated either by a fairness motive (i.e., a prosocial motive) or by the fear of an unfair offer being rejected. (For an overview of a variety of non-interdependent and interdependent tasks and their payoff structures, see Fehr & Schmidt¹⁹; Kelley et al.²⁰). In interdependent settings, the choice whether to act prosocially or not also entails strategic considerations and beliefs about the other person's response²⁰⁻²². The effects of hunger on decisions in interdependent settings could, therefore, be the result of hunger effects on prosociality but also of hunger effects on strategic decision making. We have argued previously that humans can compensate for their own psycho-physiological impairments when the social context requires them to²³. Hence, it is possible that, even if hunger enhances selfish tendencies, those will not necessarily translate into decreased prosociality in interdependent settings due to strategic concerns that also influence the decision."

3. You write that you stick to the use of Bonferroni correction. I agree that this is the proper way to test the hypothesis. Nevertheless, this weakens the evidence for the null result. Since the procedure is transparent, the reader can form his or her own opinion. Nevertheless, I would for example suggest removing the word "any" in the sentence "we did not find any significant effects of hunger on prosociality in the individual DVs".

Thank you for pointing this out. We agree, and have deleted the word "any" from the sentence in the abstract (and from other parts of the manuscript).

4. The trust game was excluded from the main analysis. In my view, it should be treated as the ultimatum game and included in the interdependent settings.

Thank you very much for your feedback on our main analysis. In Study 2, we did not use a trust game, but measured trust-in-others and trust-in-self using a self-report measure adopted from Van Lange, Vinkhuyzen and Posthuma (2014; e.g., "I completely trust most other people" / "I think that most other people trust me"; 3 items for trust-in-others and 3 items for trust-in-self, 7-point Likert scales). In our opinion, self-reported trust does not reflect a measure of prosociality but rather was included to safeguard possible effects of hunger in our interdependent settings (e.g., decreased contributions in the PGG could also be explained through decreased trust-in-others instead of increased selfishness). Moreover, due to the self-report measurement using a likert scale it is very different used from the economic games. As we did not find significant effects on the prosociality measures, we decided to report the self-reported trust measure only as an exploratory measure in the methods section for reasons of full transparency. However, we would like to point out that hunger also had no significant effect on trust:

Hungry participants reported a similar amount of trust in others ($M = 4.07$, $SD = 1.43$) as did non-hungry participants ($M = 3.95$, $SD = 1.24$), $F(1, 101) = 0.19$, $p = .661$, $d = -0.09$. Hunger also did not affect beliefs in others' trust in the self ($M_{Hunger} = 5.03$, $SD_{Hunger} = 0.86$; $M_{control} = 5.01$, $SD_{control} = 0.79$), $F(1, 101) = 0.01$, $p = .907$, $d = -0.01$.

Hence, the finding is consistent with our results.

5. Concerning deception, I appreciate that there were recipients in the dictator games. This was my main concern because otherwise participants could feel deceived because the transfer was not made. However, I would like to note that the argument that deception would only affect future experiments is quite dangerous. Actually, it implies that it would be much more efficient to block this experiment than all future ones.

Thank you very much for this pointer. We appreciate your advice and we will consider it in our future studies.

Reviewer #3 (Remarks to the Author):

For full transparency, the title should include the word "acute."

Thank you for pointing this out. We have changed the title accordingly.

Regarding the counting the number of risking choices issue, the fact that someone else took this flawed approach and got it published, is not justification for continuing the flawed approach.

Thank you for discussing the risky choice analysis with us. The main purpose of the non-social risk task was to safeguard the specificity of a potential effect of hunger on prosociality. We had included the non-social risk task to rule out an alternative explanation that hunger affects risk preferences in general (i.e., also in non-social context). However, as we found robust effects of hunger on prosociality, the non-social risk task became a bit pointless (and we omitted it from the follow-up studies 3-4). We, therefore, decided not to report the non-social risk task in our revised manuscript but only announced that we included it.

Acute hunger had no effect on the non-social risk task, independent of whether we used the approach proposed by von Dawans et al. (2012) or a more adequate approach as suggested by you.

I am still not sold that the paper is really telling us anything. I still believe the null result is because the treatment is just too weak to have any significant effect on behavior.